



# Insights into organic-aerosol sources via a novel laser-desorption/ionization mass spectrometry technique applied to one year of PM₁₀ samples from nine sites in central Europe

Kaspar R. Daellenbach[1], Imad El-Haddad[1], Lassi Karvonen[1], Athanasia Vlachou[1], Joel C. Corbin[1], Jay G. Slowik[1], Maarten F. Heringa[1], Emily A. Bruns[1], Samuel M. Luedin[2,*], Jean-Luc Jaffrezo[3], Sönke Szidat[4], Andrea Piazzalunga[5,**], Raquel Gonzalez[6], Paola Fermo[6], Valentin Pflueger[2], Guido Vogel[2], Urs Baltensperger[1], André S. H. Prévôt[1]

[1]Laboratory of Atmospheric Chemistry, Paul Scherrer Institute, CH-5232 Villigen, Switzerland
[2]MABRITEC AG, Riehen, Switzerland
[3]Université Grenoble Alpes, CNRS, IGE, 38000 Grenoble, France
[4]Department of Chemistry and Biochemistry & Oeschger Centre for Climate Change Research, University of Bern, 3012 Bern, Switzerland
[5]Università degli Studi di Milano-Bicocca, 20126 Milano, Italy
[6]Università degli Studi di Milano, 20133 Milano, Italy
[*]now at: University of Geneva, CH-1211 Geneva, Switzerland
[**]now at: water and soil lab, 24060 Entratico, Italy

*Correspondence to*: André S. H. Prévôt (andre.prevot@psi.ch)

**Abstract.** We assess the benefits of offline laser-desorption/ionization mass spectrometry (LDI-MS) in understanding ambient particulate matter (PM) sources. The technique was optimized for measuring PM collected on quartz-fiber filters using silver nitrate as an internal standard for $m/z$ calibration. This is the first application of this technique to samples collected at nine sites in central Europe throughout the entire year 2013 (819 samples). Different PM sources were identified by positive matrix factorization (PMF) including also concomitant measurements (such as $NO_x$, levoglucosan, and temperature). By comparison to reference mass spectral signatures from laboratory wood burning experiments as well as samples from a traffic tunnel, three biomass-burning factors and two traffic factors were identified. The wood-burning factors could be linked to the burning conditions; the factors related to inefficient burns had a larger impact on air quality in southern Alpine valleys than in northern Switzerland. The traffic factors were identified as primary tailpipe exhaust and most possibly aged/secondary traffic emissions, respectively. The latter attribution was supported by radiocarbon analyses of both the organic and elemental carbon. Besides these sources, also factors related to secondary organic aerosol were separated. The contribution of the wood burning emissions based on LDI-PMF correlates well with that based on AMS-PMF analyses, while the comparison between the two techniques for other components is more complex.

## 1 Introduction

Climate and health are strongly affected by atmospheric aerosols (Kelly et al., 2007; IPCC, 2013), a substantial fraction of which is organic (Jimenez et al., 2009 and reference therein). This organic aerosol (OA) is a complex mixture of thousands of compounds (Goldstein and Galbally, 2007), of which only 10-30% have been speciated by modern techniques (Hoffmann et al., 2011; Simoneit et al., 2004). Therefore, the chemical composition of OA, its emission sources and formation processes are under ongoing investigation. OA can be directly emitted as primary particulate matter (primary OA, POA) or formed through the oxidation of gas-phase precursors with subsequent condensation or nucleation (secondary OA, SOA). SOA dominates submicron OA at remote sites (90%) and is also a substantial contributor (30-80%) in the urban environment (Zhang et al., 2011).



Mass spectrometry has significantly advanced the chemical characterization and quantification of OA. Instruments equipped with electron-impact (EI) ionization, such as the aerosol mass spectrometer (AMS, Canagaratna et al., 2007) and aerosol chemical speciation monitor (ACSM, both Aerodyne Research, Inc., Ng et al., 2011a; Fröhlich et al., 2013) provide quantitative online measurements of OA (Jimenez et al., 2016). The application of positive matrix factorization (PMF) to

AMS and ACSM mass spectra has allowed the separation of different POA sources such as traffic, cooking, and wood burning, as well as oxygenated OA factors representing SOA (e.g. Jimenez et al., 2009; Lanz et al., 2007; Lanz et al., 2008; Lanz et al., 2010; Crippa et al., 2014, Canonaco et al., 2013). While different SOA factors identified by AMS-PMF have been separated according to their degrees of oxygenation and volatility, information on the different origins of this fraction is limited due to significant fragmentation of the molecules by EI. Moreover, SOA from different sources converges to a

chemically similar composition during oxidation (Kroll et al., 2011, Ng et al., 2011b).

Other strategies for aerosol mass spectrometry have provided complementary information. Ionization by electrospray (ESI) avoids significant analyte fragmentation. The coupling of ESI to ultra-high-resolution Fourier-transform mass spectrometers thus provides detailed information on the chemical composition of a sample. However, using ESI not all compound classes

can be detected efficiently due to ion suppression (Furey et al., 2013; Trufelli et al., 2010; Kourtchev et al., 2013). In addition, high costs and labor intensity restrict the application typically to smaller sets of samples. Traditional techniques such as gas- or liquid-chromatography (Hoffmann et al, 2011) coupled to MS suffer from similar restrictions.

Laser-based mass spectrometers such as laser-desorption/ionization mass spectrometers (LDI-MS) and two-step laser mass

spectrometers (L2MS) are, similarly as ESI, less affected by fragmentation than EI. Haefliger et al. (2000a and 2000b) show that such instrumentation allows for identification of the dominant primary sources using mass spectral fingerprints (tracer $m/z$'s identified using emission samples and principle component analysis of ambient measurements). LDI-MS instruments may be subject to matrix effects, i.e. ion formation from a given compound does not only depend on its abundance but also on the abundance of all other compounds (De Hoffmann and Stroobant, 1999). Additional matrix sensitivities may arise if

ion-neutral reactions take place in the desorbed plume, which means that MS design (e.g. time-of-flight vs. ion trap) also plays a role (Murphy, 2007). Although the importance of matrix effects is evident, studies with a single-particle LDI-MS (ATOFMS, laser wavelength 266 nm) have achieved good correlations of major aerosol components with well-established reference measurements like elemental carbon (EC, Sunset Analyzer, Sunset Laboratory Inc.), OA (AMS), $NH_4$ (AMS), $SO_4$ (AMS), $NO_3$ (AMS), and K (collected with a particle-into-liquid sampler, PILS), suggesting that underlying matrix effects

did not dominate measurement reproducibility (Healy et al., 2013).

Online single-particle LDI-MS instruments, like the ATOFMS, generally employ very high laser fluences. This leads to similar fragmentation as observed in the AMS. However, such fragmentation can be avoided. By measuring offline aerosol samples (filters) at different laser powers (LDI-MS, Shimadzu/Kratos, Axima CFR) and with and without adding a matrix,

Samburova et al. (2005a) showed that fragmentation was negligible in their instrument. Overall, offline LDI-MS may therefore provide quick access to additional chemical information at near-to-molecular level, potentially allowing differentiation between several primary organic aerosol sources and even different precursor-related SOA categories (Kalberer et al., 2004; Samburova et al., 2005a). Based on offline LDI-MS measurements performed for an urban background site in Zurich, Switzerland (same site as Haefliger et al., 2000a; 2000b), Baltensperger et al. (2005) suggested

that SOA from biogenic precursors is more important than SOA from anthropogenic precursors. However, such LDI-MS analyses are rare, typically focusing on pattern analysis in the mass spectrum since LDI-MS signal quantification is difficult due to the variability in ionization efficiencies and chemistry for different compounds. In addition, LDI-MS has not been





applied to extensive datasets. In comparison to online analyses, analyzing offline filter samples allows covering longer time periods and larger observation networks.

In this work, we evaluated the use of offline LDI-MS for the direct measurements of PM collected on filters, without the addition of an ionization matrix. We assessed the contributions of different aerosol sources to the total PM. To control for instrumental differences, we measured all filters on the same instrument. We developed novel procedures for instrument calibration and uncertainty assessment. We measured three source reference samples (traffic, wood burning, and cooking) in addition to 819 ambient samples. The ambient samples included filters from the entire year of 2013 at nine sites in central Europe with different emission conditions (Alpine valleys strongly influenced by wood burning, as well as urban and rural regions). Based on positive matrix factorization (PMF) using LDI-MS mass spectral data, the ability to resolve OA sources in  source apportionment was assessed and used for obtaining a deeper understanding of sources contributing to the organic aerosol in central Europe.

## 2. Methods

### 2.1 Sample collection and other chemical analyses

Ambient samples were collected at nine sites in Switzerland and Liechtenstein (described in more detail in Daellenbach et al., 2017) covering different atmospheric conditions (urban/rural and background/curbside). Seven sites were located on a plateau north of the Alpine crest (Basel, Bern, Payerne, Frauenfeld, St. Gallen, Zurich, Vaduz); the remaining two sites were located in Alpine valleys south of the Alps (Magadino, San Vittore). Samples were collected on quartz filters (Pall Corp.) by local air quality monitoring networks every $4^{th}$ day during the entire year 2013 (819 filters) and used for gravimetric $PM_{10}$ quantification. The samples were stored at -18°C and transported in cooling boxes. Before any further handling steps, the samples were allowed to warm up at room temperature for 60 minutes in order to avoid condensation of ambient humidity. The organic (OC) and elemental (EC) carbon content was determined by a thermo-optical transmission method using a Sunset OC/EC analyzer (Sunset Laboratory Inc., Birch and Cary, 1996), following the EUSAAR-2 thermal-optical transmission protocol (Cavalli et al., 2010). For 33 samples from Magadino, radiocarbon ($^{14}$C) analyses were performed (2013; 10 samples, 2014; 23 samples, Vlachou et al., in prep). The radiocarbon analyses were conducted at the University of Bern at the Laboratory for the Analysis of Radiocarbon with the Accelerator MS (LARA; Szidat et al., 2014) to determine the contribution of fossil and non-fossil OC ($OC_f$ and $OC_{nf}$, respectively) and EC ($EC_f$ and $EC_{nf}$, respectively) content (Zhang et al., 2012). Major ion concentrations were measured by ion chromatography (IC) by a Dionex ICS1000 instrument (Piazzalunga et al., 2013; Jaffrezo et al., 1998). Levoglucosan was measured by high-performance anion exchange chromatography (HPAEC) with pulsed amperometric detection (PAD) using an ion chromatograph (Dionex ICS1000) following the method of Piazzalunga et al. (2010; 2013a). Gas-phase $NO_x$ was measured online using a chemiluminescence method, and meteorological parameters (e.g., temperature) were monitored. Further, equivalent black carbon (eBC) was measured with a multi-wavelength Aethalometer AE 31 (Magee Scientific Inc.) (Hansen et al., 1984; Herich et al., 2011) in Magadino, Payerne and Zurich. eBC was separated into a wood burning influenced ($eBC_{wb}$) and traffic influenced ($eBC_{tr}$) fraction based on the enhanced absorption of $eBC_{wb}$ in the ultraviolet range. For this computation, we used the Ångström exponents for wood-burning ($\alpha_{wb}$) of 1.7 and for traffic ($\alpha_{tr}$) 0.9 (Zotter et al., 2017; Sandradewi et al., 2008). On all the samples, offline AMS analyses (Daellenbach et al., 2016) were conducted. This involved the analysis of the water-soluble organic matter (WSOM) by a high resolution time-of-flight AMS. The offline AMS data were used for source apportionment (Daellenbach et al., 2017; Bozzetti et al., 2016) and used in this study for comparison; the source mass concentrations were corrected for the missing water-insoluble mass fraction using the method described in Daellenbach et al. (2016).





Reference source samples for traffic (representing $PM_{10}$) were collected in a tunnel (Islisberg tunnel on Swiss highway A4, exit Wettswil am Albis). The $PM_{10}$ samples were collected on Sunday, 2014-05-18, and Tuesday, 2014-05-20, from midnight to noon with 2 hour intervals (rush hour in the morning). Additional source samples were collected on quartz filters for beech-wood burning through laboratory experiments of whole cycle burns and stable flaming phase only. These samples

did not only include the primary aerosol, but also two different levels of aging, i.e., after simulated aging in a smog chamber for 1 hour (equivalent to an OH dose of $10^7$ cm$^{-3}$ h ) and 4 hours (OH dose $3 \cdot 10^7$ cm$^{-3}$) (Bruns et al., 2015; 2016). Additionally, also filter samples collected during cooking experiments were analyzed (Klein et al., 2016a, 2016b).

**2.2 Laser-desorption/ionization ToF MS analysis**

**2.2.1 Instrument and measurement settings**

In laser desorption/ionization mass spectrometry, LDI-MS, liquid or solid material is simultaneously desorbed and ionized by pulsed laser irradiation. The laser is focused on the surface of the sample. Ions are produced through two major pathways (Knochenmuss et al., 2000; Zenobi et al., 1998). In the first pathway, ions are formed by interaction with the laser beam (primary ions). In the second pathway, in the expanding primary plume of desorbed molecules and primary ions, ion-

molecule reactions produce secondary ions (Knochenmuss et al., 2000). The detailed ionization mechanisms are still under investigation (Knochenmuss et al., 2002; 2003; 2006; 2016). Although this type of ionization is considered softer than electron ionization, the observed ions are typically still fragments of their parent molecules (De Hoffmann and Stroobant, 1999). Furthermore, the measured intensity of one compound does not only depend on its concentration but also on the concentration of all other compounds present (the so-called matrix effect) making quantification challenging (Ellis et al.,

2014; Borisov et al., 2013).

We recorded the mass spectra of 819 filter samples (*m/z* range 65-500, ion gate at *m/z* 60) using a laser-desorption/ionization-ToF MS (Shimadzu Axima Confidence, Shimadzu-Biotech Corp., Kyoto, Japan) equipped with an $N_2$ laser (wavelength 337 nm, frequency 50 Hz, laser pulse width 3 ns) in the positive reflectron mode. All the accessible

instrumental parameters were kept constant during the whole period of measurements taking place from November 2015 to mid-March 2016. Specifically, the laser intensity was adjustable by means of a rotating wheel of filters with varying transmission. While this wheel was initially set and kept constant, the aging of the laser during the given time period was also expected to reduce its intensity. We monitored and assessed changes in laser power and other instrumental parameters, as well as possible sources of uncertainty/contamination from sample preparation and intra and inter-day reproducibility, by

repeated measurements of a subset of our samples.

Quartz filter punches of 8 mm were attached to a custom-made stainless steel sample holder (32 slots). Each of the samples was additionally spiked with a droplet of dilute aqueous silver nitrate solution ($AgNO_3$, Sigma Aldrich, >99.8%, 500 ppt to 20 ppm), to provide Ag cations as an internal standard for the *m/z* calibration. After drying under ambient conditions, filter

punches were inserted into the sampling chamber and analyzed by the LDI-MS. Blanks were measured according to the same procedure. Both intra-day and delayed repeated measurements were conducted for the same filters to assess instrument performance and uncertainty/contamination from sample preparation. The intra-day repeatability was assessed by measuring 3 filters 10 times on 3 different occasions as outlined in Section 3.1. Overall, 96% of the available ambient filter samples (785 filters) provided usable data (defined below).






### 2.2.2 Data treatment

While most atmospheric LDI-MS studies present raw mass spectra (Samburova et al., 2005a; 2005b; Kalberer et al., 2004; Baltensperger et al., 2005), in the present study, we introduced data treatment techniques in order to perform further mass spectral analysis on stick-integrated spectra. The techniques we employed are described in the following.

The $m/z$ calibration was performed using silver nitrate solution (AgNO$_3$, Sigma Aldrich, >99.8%) as an internal standard (approach illustrated in Fig. 1). In order to avoid the suppression of the sample signal, the internal standard (aqueous solution 500 ppt to 20 ppm) was only placed (as a droplet) on a small part of the sample. The 499 spectra from all positions on the measurement grid were separated and defined as (1) silver-spiked (lower panel in Fig. 1a), (2) silver-free (upper panel in Fig.

1a), and (3) intermediate-silver. Intermediate-silver cases were defined using the signal intensity in the regions of the mass spectrum where silver was expected, in comparison to adjacent silver-free regions of the spectrum, and were discarded.

In an initial calibration, we calibrated the average silver-spiked spectrum of each filter sample, using the peaks of the silver monomer ($m/z$ 107, 109), dimer ($m/z$ 214, 216, 218), and trimer ($m/z$ 321, 323, 325, 327). We found that this calibration of

silver-spiked was not directly applicable to the average silver-free mass spectrum. Possibly, spiking the filter region with aqueous AgNO$_3$ caused enough of a change to the surface of the filter sample to influence the ionization physics. This could affect the $m/z$ calibration, since ion extraction in our instrument is not orthogonal to the ablation plume. Therefore, the averaged silver-free spectra were recalibrated with respect to the silver-spiked spectra using prominent non-silver peaks present in both spectra. Such a two-step calibration is necessary to achieve accurate $m/z$ calibrations of the silver-free

spectra.

After calibration, mass spectra were baselined using the following custom algorithm. A window of width 1 u (u being the unified atomic mass unit, the detector bins were approximately 0.04 u in width) was applied to identify the lowest signal intensity of that range, which was defined as the baseline. The window was moved across the mass spectrum with steps of

0.25 u to obtain a baseline spectrum. After linear interpolation, the baseline was subtracted from the spectrum. Subsequently the spectra were integrated to unit mass resolution sticks (UMR, 1 stick per unit mass). The UMR integration window was not centered at integer masses, but rather at integer mass plus the mass defect of an alkyl ion (R-CH$_2^+$). The width of this window was defined as extending to the minimum signal intensity above and below the center using the first derivative of the signal.


To enable a direct comparison of LDI-MS and HR-AMS results, the intensity scales of the calibrated and baseline-subtracted spectra ($I'_{m/z}$) with arbitrary units were scaled to OM (μg/m$^3$) using the respective OM contents of the filter samples (see Eq. 1, with rescaled intensities termed $I_{m/z}$ thereafter). OC$_{Sunset}$ was determined using the Sunset OC/EC analyzer and (WSOM/WSOC)$_{oAMS}$ through WSOM measurements (described in Section 2.1). Note that an additional contribution of non-

organic material to the signal could not be fully excluded. However, we consider the detected signal mainly to be related to organics based on the measurements of ambient aerosol with offline LDI-MS systems using different laser-wavelengths (193 nm and 355 nm, Aubriet et al., 2010) and pure components (337 nm, Goheen et al., 1997) indicating that sulfate and nitrate respond only in the negative mode. Additionally, molecular nitrate and ammonium ions are too small to be detected in this study (cutoff at $m/z$ 65). We also do not expect clusters between inorganic cations and organics because we did not observe

significant amounts of silver/organic clusters in silver-spiked spectra (Fig. 1a).

$$I_{m/z} = \frac{I'_{m/z}}{\sum_{m/z} I_{m/z}} * OC_{sunset} * \left(\frac{WSOM}{WSOC}\right)_{oAMS} \tag{1}$$





### 2.3 Source apportionment / PMF

#### 2.3.1 General and input data

Positive matrix factorization (PMF, Paatero and Tapper, 1994; Paatero, 1997) is a widely used algorithm for source

apportionment. PMF (Eq. 2) explains the variability in a dataset ($x_{i,j}$, here LDI-MS mass spectra scaled to OM, $I_{m/z}$) as a linear combination of constant factor profiles ($f_{k,j}$, here mass spectral signatures) and their time-dependent contributions ($g_{i,k}$, here concentrations of this factor). The index $i$ represents a specific point in time, $j$ the signal at a specific $m/z$, and $k$ a factor (up to the number of factors $p$).

$$x_{i,j} = \sum_{k=1}^{p} g_{i,k} f_{k,j} + e_{i,j} \qquad\qquad\qquad (2)$$

PMF is solved by minimizing the residuals $e_{i,j}$ weighed by the measurement uncertainty, here $\sigma_{PMF,i,j}$. In this study, PMF was solved by the multilinear-engine 2 (Paatero, 1999 and references therein) using the front-end user interface SoFi 4.9 (Canonaco et al., 2013) developed for Igor Pro v6 (Wavemetrics). We performed PMF without constraints of e.g. reference

mass spectra of aerosol sources (unconstrained PMF). The data matrix consisted of the stick-integrated LDI-MS spectra ($m/z$ 65 to 485 scaled to OM) of all 785 filter samples. We excluded all $m/z$'s for which we expected silver signals (listed in section 2.2.2) even though only silver-free spectra (according to the definition given above) were considered.

We determined the PMF uncertainty matrix ($\sigma_{PMF,i,j}$) based on the instrument repeatability (described in Section 3.1)

according to the two-component model for measurement uncertainty described by Rocke and Lorenzato (1995) and Wilson et al. (2004). This approach ultimately means that an absolute-error term (e.g., due to noise) is combined in quadrature with a relative-error term (e.g., due to scaling or calibration uncerties) as shown Fig. 2 and in Eq. 3 ($\sigma'_{PMF,m/z}$, for a single mass spectrum). The need for and applicability of such an error model has been demonstrated by Corbin et al. (2015) for mass-spectrometry–PMF. We obtained the two error terms by fitting Eq. 3 to the relative standard deviation of replicate

measurements for each filter and experiment, and applied the averaged fitted parameters to the entire data set (constant error term, $\sigma_{abs}$, and an error term proportional to the measured peak intensity, $I'_{m/z}\sigma_{rel}$).

$$\sigma'_{\mathrm{PMF},m/z} = \frac{\sqrt{\sigma_{\mathrm{abs}}^2 + I'^2_{m/z}\sigma_{\mathrm{rel}}^2}}{I'_{m/z}} \qquad\qquad\qquad (3)$$

The error was scaled to OM in the same way as the LDI-MS mass spectra (Eq. 4).

$$\sigma_{\mathrm{PMF},m/z} = \sigma'_{\mathrm{PMF},m/z}\left(\frac{I_{m/z}}{I'_{m/z}}\right) \qquad\qquad\qquad (4)$$

#### 2.3.2 Uncertainty estimate of PMF results

A well-established statistical tool for estimating the uncertainty is the bootstrap technique, which consists of randomly

resampling the input data, with replacement, to create input matrices with the same dimensions as the initial data matrix (Davison and Hinkley, 1997; Brown et al., 2015). We performed 1'000 bootstrap PMF runs (also using the replicate measurements) with different initial guesses ("seeds") using input matrices of the same size (785 filter samples and 412 ions), resulting in the uncertainty estimate $\sigma_{bs}$. We also assessed and parameterized the uncertainty arising from the intra-day





repeatability for three filters analyzed repeatedly by LDI-MS on 3 different days, 10 times each day ($\sigma_{intraday}$, see SI). The reported uncertainty for each PMF factor $\sigma_{tot}$ is the quadratic sum of $\sigma_{bs}$ and $\sigma_{intraday}$, as explained in more detail in the Supplement. Potential long-term drifts in instrumental response were evaluated ($\sigma_{interday}$, see Supplement, Fig. S10) but were not accounted for numerically.

## 3. Results and Discussion

### 3.1 Calibration, repeatability and quantification

An example of a sample spectrum after $m/z$ calibration and baseline subtraction, separated into silver-spiked and silver-free spectra and averaged over the entire filter punch, is presented in Fig. 1a. The silver-spiked spectrum features an intensive signal at $m/z = 214$ related to the $^{107}Ag_2$ dimer, in contrast to the silver-free spectrum.

The operational mass resolution, $m/\Delta m$, based on the measurements conducted on one sample tray per month and the accuracy of both calibration steps are shown in Fig. 1b and 1c, respectively. The operational resolution of the instrument was only determined for the silver mono- (resolution 1100), di- (1700), and trimer (2100) as we are confident of the absence of interfering ions for these peaks. In comparison to the LDI-MS, the HR-ToF AMS in V-mode (and W-mode) has a resolution of ca. 2'000 (4'000) at $m/z$ 100 and an $m/z$ calibration accuracy < 20 ppm (< 10 ppm, DeCarlo et al., 2006). With this resolution, distinguishing different ions at the same nominal mass was not possible for the quartz-filter-LDI-MS measurements. Instead, the spectra were integrated to UMR sticks.

The repeatability of the mass spectral signature was assessed in detail for 3 selected filters (collected in Basel on 2013-06-21, 2013-09-21, and 2013-12-22). Ten punches from each of these 3 samples were prepared on the sample holder in the same time and subsequently measured. The given procedure was repeated on 3 occasions (on 2015-11-25, 2015-12-17, 2016-01-28). The relative error was related to the absolute signal intensity for each $m/z$ for each sample and experiment (Fig. 2), with an asymptotic value of only ~9 ± 7 % at high signals (average and standard deviation of fitting Eq. 3). The average absolute error for small signals was 102 ± 48 a.u. (from Eq. 3). These parameters did not show a temporal trend which suggests that the measurements are repeatable, despite filter inhomogeneity and laser instability. Analyses of field blanks (also spiked with $AgNO_3$) exhibited low signal with 99% of peaks below detection limit (defined as $3 * \sigma_{abs}$). The other 1% of the peaks ($m/z$ 197, 249, 251, 322,324) exhibited high variability among the field blank analyses (see details in Supplement, Fig. S1 and S2). Therefore, no blank subtraction was performed.

The total measured intensity ($I_{tot} = \sum I_{m/z}$) did not show a relation with the filter loading (OC, OC+EC, $PM_{10}$, see Supplement, Fig. S3), indicating that factors such as the composition, size, and/or mixing state of the collected aerosol particles had a larger influence on the measured ion intensities than the mass of PM. Moreover, replicate measurements indicated that the $I_{tot}$ of a single sample measured on different instances was affected by an instrumental drift (details in Supplement, Fig. S4).

### 3.2 Combustion-source samples

Filters from cooking processes (frying, related largely to oil pyrolysis, Klein et al. 2016a; Allan et al., 2010) did not yield measurable mass spectra in our LDI-MS. This observation is in agreement with a prior study which performed in-situ single-particle LDI-MS measurements at 266 nm (using the ATOFMS) and did not observe cooking particles (Healy et al., 2013). The lack of either graphene-like black carbon or polycyclic aromatic hydrocarbons (PAHs) to absorb the LDI-MS laser may



explain these observations (Healy et al., 2013), which suggests that our measurements are more sensitive towards certain PM components. We speculate that it may be possible in some cases to observe cooking-related ions in LDI-MS of atmospheric samples, if such ions are secondary ions formed in the ablation plume or cooking particles are coated by absorbing SOA.

The mass spectra of tunnel filter samples were complex and characterized by distinct patterns (Fig. 3). Some $m/z$ fragments (e.g. 84, 94, 101, 177) were present in the weekday and weekend tunnel samples, but not in the biomass smoke. Between the two tunnel samples, large differences could also be identified. The tunnel sample collected on a Tuesday during rush hour (06:00-08:00) showed many more peaks (among others 163 and 177) which were less prominent during the weekend at the same time and in the same location (where $m/z$ 101 and 143 dominated the weekend spectrum).

Previous studies have identified lubrication oil as a major component of diesel- and gasoline-vehicle exhaust (Gentner et al., 2017; Chirico, et al., 2011), which consists largely of aliphatic hydrocarbons. Cooking particles similarly consist of such hydrocarbons (Schauer et al., 2002; Allan et al., 2010; Crippa et al., 2013). Since we did not observe a mass spectrum for the cooking sample, but do observe one for the tunnel sample, our detection of tunnel particles may rely either on the presence

of black carbon or PAHs on the filter samples. To the extent that these species are heterogeneously distributed within the sampled aerosol particles, our technique may be specifically sensitive to certain components of traffic emissions.

The mass spectrum of the wood burning sample from the whole burning cycle shows a complex pattern with a bimodal envelope of relatively-intense ions from $m/z$ 200 onwards. By contrast, the sample from the stable-flaming phase shows

fewer peaks (e.g. $m/z$ 85, 140, 213, etc.) with a less significant envelope of ions.

### 3.3 Ambient samples

Typical mass spectra from 4 different locations, known to be heavily influenced by specific sources at different times of the year are displayed in Figure 4. The fragments are color-coded by their correlation coefficients with 3 different markers or proxies for aerosol types, i.e., $NO_x$ for traffic exhaust (Fig. 4a), levoglucosan for biomass burning emissions (Fig. 4c and 4d)

and temperature for SOA production (as temperature is expected to exponentially enhance biogenic emissions, Leaitch et al., 2011, Fig. 4b). In general, the mass spectra from winter-time were characterized by higher contributions from higher-molecular-weight fragments compared to summer. For the winter sample from San Vittore, a large "hump" was present, on top of which signals appeared with high intensities and a highly regular pattern of $m/z$ differences of 14 Th. These fragments, albeit not all detected in smog chamber biomass burning aerosols, were highly correlated with levoglucosan, a unique

marker of biomass smoke.

The $m/z$ 84 94, 101, 120, 143, 165, and 177, also detected in tunnel samples, showed high relative intensities in Bern in summer, of which 84, 94, 120, and 177 correlated with $NO_x$, which suggests a high contribution of traffic emissions at this location, consistent with previous observations (Zotter et al., 2014). These ions were also observed, but to a lesser extent, at

the rural site of Payerne. At the latter site, the highest intensities were found for $m/z$ 74 and 104. Only these 2 ions showed a clear relation with temperature and a clear increase during summer. Overall, this spatial and temporal variability observed in the spectral fingerprints and the correlation of specific fragments with environmental parameters suggest that LDI-MS data contain mass spectral information that may be used to separate the contribution from different aerosol sources. Source apportionment analysis using these data in combination with PMF was therefore explored in the following.






### 3.4 Source apportionment results

#### 3.4.1 PMF setup

Residual analysis of preliminary PMF runs showed that structure in the residuals was removed when increasing the number of factors up to 5 but not when further increased. However, when allowing for 7 factors, further environmentally

interpretable separations could be achieved. Increasing to 8 factors led to the separation of a third traffic related factor, which however contributed little to the overall mass. Therefore, we opted for 7 factors (see details in Supplement, Fig. S6, S7, and S8).

#### 3.4.2 Interpretation of PMF factors

**3.4.2.1 Traffic-related factors**

Two factors could be related to traffic (Fig. 5a and 5b, termed traffic1 and traffic2, respectively) based on patterns in the factor profiles similar to patterns in the samples measured in the tunnel on a weekday (Fig. 3a). Since the signatures obtained from the tunnel filters represent both tailpipe exhaust and resuspended dust, the two factors identified could be a mixture of both sources. In order to elucidate the reasons behind the separation of traffic related aerosols into two factors by PMF, we

inspected their relationship with $NO_x$ and $eBC_{tr}$, typical markers of traffic emissions. Both traffic1 and traffic2 showed a relationship to $NO_x$, only traffic2 correlated with $eBC_{tr}$ (Tab. S1). This suggests that traffic2 is related to primary emissions from the combustion process and that traffic2 is also influenced by other processes, like e.g. secondary production. Additionally, tailpipe exhaust cannot be distinguished from other traffic related emissions in the tunnel samples and might also contribute to traffic1.

The ratio of both factors to $NO_x$ exhibited a clear seasonality, increasing during summer (Fig. 6a and 6b). Such a change can either be caused by a change in the emission patterns, by an enhanced photochemical production of these factors or due to a change in the lifetime of e.g. $NO_x$. In Fig. 6c and d, we compared the contribution of the traffic related factors with those of $eBC_{tr}$, another tracer of traffic emissions whose lifetime is similar to OA (see Fig. S9 for summer points only). The

comparison shows that while the ratio of traffic2 to $eBC_{tr}$ is not season-dependent, the one of traffic1 to $eBC_{tr}$ increases during summer. The seasonal variability in the relative contribution of traffic1 might thus be related to a seasonal change in fleet composition or combustion conditions, or an enhancement of the photochemical production of this fraction. An enhanced dust resuspension in the warm season (e.g., more dust on the road due to less precipitation) could also contribute to an increased traffic1 concentration in summer.


#### 3.4.2.2 Wood-burning-related factors

Three factors could be related to wood burning emissions (Fig. 5c, 5d, 5e). The first (Fig. 5c) showed a similar mass spectral pattern as samples from laboratory experiments from stable flaming wood burning exhaust from a log wood burner and thus was termed efficient wood burning (BBeff). The fingerprint of the second (Fig. 5d) resembled that of laboratory wood smoke

aerosols from the entire burning cycle including the inefficient starting and burnout phase and, therefore, was related to inefficient wood burning (BBineff1). BBineff1 showed also a similar signature as spectra from wood burning haze episodes in San Vittore identified by high levoglucosan concentrations. The third (Fig. 5e) mostly explained the masses above *m/z* 300. The signature of the whole cycle wood burning aerosols from a log wood burner (Fig. 3c) also showed high relative contributions at high *m/z*, similar to this factor. For this reason this factor was termed BBineff2. Both BBeff and BBineff1

correlated with levoglucosan (Fig. 6e, 6f, Tab. S1). Similar to BBeff and BBineff1, BBineff2 correlated with levoglucosan



and thus could be related to wood burning emissions (Fig. 6.g, Tab. S1). However, BBineff2 also correlated with $NH_4^+$, a marker of aged aerosols, at the northern sites (Fig. 6m), which may also indicate that this fraction can include aged and/or secondary components (Tab. S1).

Comparing the wood burning related factor time series to potassium ($K^+$), an inorganic wood burning marker mostly present in ash, provides further insight into the separation of the 3 wood burning related factors. Among the factors related to wood burning emissions, BBeff has a lower BB/$K^+$ ratio (2.4, IQR 1.25-4.2) than BBineff1 (6.0, IQR 2.7-12.0) and BBineff2 (11.6, IQR 6.0-21.0). In wood burning experiments, it was found that OC, EC, $PM_{10}$, and PAH emissions increase relative to the potassium output during non-ideal burning conditions (Lamberg et al., 2011). Zotter et al. (2014) found a north-south

gradient in levoglucosan/$K^+$ and $OC_{nf}$/$K^+$ in Switzerland and hypothesized that it might be linked to the burning conditions. Thus the higher the LDI-BB/$K^+$ ratios (north: 16.6, IQR 10.4-30.8 south: 30.8, IQR 22.5-44.2), the less efficient the burning conditions which in turn supports the hypothesis that BBeff represents the most efficient burning conditions among the three factors. All wood burning related factors showed higher median BB/$K^+$ ratios at the southern Alpine valley site (BBeff: 3.0 with IQR 1.1-5.2, BBineff1: 10.9 with IQR 7.5-17.3, BBineff2: 15.8 with IQR 9.9-24.0) than in northern Switzerland

(BBeff: 2.1 with IQR 1.2-3.4, BBineff1: 4.3 with IQR 2.2-8.6, BBineff2: 9.6 with IQR 5.2-18.3) as also visible in Fig. 6h/i/j. The north-south gradient in the BB/$K^+$ ratios might be caused by imperfections of the separation of the burning conditions by PMF, but other effects like the age of the stove population and the used technology could contribute as well. The wood consumption (BFE, 2013) of automatic burners (>50kW), was higher in northern Switzerland (in the respective regions, 48$m^3$ wood/$km^2$ area) than in southern Switzerland (8$m^3$ wood/$km^2$ area). Since the OM, POA, and SOA emissions of pellet

burners during the stable phase are drastically reduced compared to modern logwood burners in a stable flaming phase (e.g. Heringa et al., 2011), they might contribute over-proportionally to potassium in northern Switzerland but only little to OM, leading to the higher BB/$K^+$ ratios at the southern Alpine sites.

### 3.4.2.3 Biogenic-SOA, low-molecular-weight OA, and other OA factors

A factor characterized by high contributions of low-molecular weight ions (Fig. 5f, LMW-OA) correlates with $NH_4^+$ suggesting secondary processes as origin (Fig. 6l, Tab. S1). At the southern Alpine valley sites, the LMW-OA / $NH_4^+$ ratio was higher than in northern Switzerland. As $NH_4^+$ is mostly associated with the secondary inorganic species sulfate and nitrate this variability will mostly be related to differences in the VOC versus $SO_2$ and $NO_x$ emissions, along with temperature differences influencing the partitioning of nitrate to the particle phase.


The contribution of the last remaining factor showed an exponential increase with temperature, similar to terpene emissions and biogenic SOA (Leaitch et al., 2011), suggesting this factor to be strongly influenced by biogenic SOA production. Therefore, this factor was termed bio-OA. The relationship between bio-OA and temperature was similar both at the northern and southern sites (Fig. 6k, Tab. S1). The chemical signature of bio-OA was dominated by fragments at *m/z* 74 and

104 (Fig. 5g). The nature of these fragments remained unidentified, but could not be related to mono-terpene or sesquiterpene SOA because of their very low *m/z*. Further, the composition of secondary aerosols is expected to be much more complex, showing a series of fragments distributed over a wide *m/z* range. We note that the abundance of the fragments at *m/z* 74 and 104 relative to the total signal should not be directly related to their absolute concentrations and their ionization efficiency might be superior to that of other molecules, which may increase their apparent contribution.





### 3.4.3 Comparison to offline AMS results and earlier AMS campaigns

The LDI-MS source apportionment results were compared to those based on offline AMS (oAMS) analyses performed on the same samples (Daellenbach et al., 2017). We note that this comparison is not straightforward as different sources are separated by the two methods. For this purpose traffic1 and traffic2 were summed up to LDI-traffic, and BBeff, BBineff1,

and BBineff2 to LDI-BB (Fig. 7).

As was the case for the comparison to $NO_x$ and $eBC_{tr}$ (Section 3.4.2), also in the comparison to the offline AMS traffic (HOA), a higher LDI-traffic/HOA was observed in summer which contributes to the low correlation coefficient ($R^2$=0.05). This might be related to the fact that HOA is only primary without major contributions of traffic SOA or dust resuspension

while LDI-traffic is potentially also influenced by aged/secondary traffic aerosol and resuspension. In contrast to summer, we observe a good agreement between LDI-traffic and HOA in winter. Thus even though a varying response factor of the LDI-MS might contribute to these differences, these biases are not systematic but season dependent. As stated earlier, LDI-traffic is thought to be a mixture of primary tailpipe exhaust, aged/secondary tailpipe exhaust, and resuspended dust (as well as tyre break and engine wear). This will be further elucidated in Section 3.4.5.

LDI-BB was highly correlated with offline AMS-BBOA ($R^2$=0.82), yet the LDI-BB concentrations were higher than BBOA from oAMS, especially in northern Switzerland (with an LDI-BB:AMS-BBOA ratio between 1.2 and 4.2 for the different sites). A possible reason is also here the mixing of secondary components into LDI-BB when comparing to primary BBOA from oAMS. However, we cannot exclude that this effect is due to different ionization efficiencies of the LDI-MS for

different compound classes.

The identified secondary components, LMW-OA ($R^2$=0.47) and bio-OA ($R^2$=0.61), also correlate with the corresponding OOA factors from the oAMS analysis, WOOA and SOOA, yet the correlation coefficients are smaller than for LDI-BB and BBOA. For these factors, differences between the two methodologies could be related to differences in the PMF

performance or to differences in the response factors for different components in the LDI (the LMW-OA: WOOA ratio is between 0.6 and 1.0 and the bio-OA : SOOA ratio between 0.1 and 0.4).

At some of the sites analyzed with LDI-MS in this study, OA was monitored in previous years with state-of-the-art online aerosol mass spectrometry (AMS or ACSM). Earlier campaigns with quantitative online AMS analyses show a higher SOA

contribution at those sites: e.g. in Zurich, July 2005 (66% vs 25% for LDI-MS summer) and December 2005 (55% vs 14% for LDI-MS winter), in Roveredo (close to San Vittore), in March 2005 (53% vs 46% for LDI-MS summer), and in December 2005 (43% vs 11% for LDI-MS winter), in Payerne, in July 2005 (94% vs 47% for LDI-MS summer) and in December 2005 (71% vs 20% for LDI-MS winter) (Lanz et al., 2010). Canonaco et al. (2013) presented source apportionment results for the winter of 2011 and 2012 for Zurich ($PM_1$ ACSM), also with higher SOA contributions (71%)

than the LDI-MS in 2013 (20%).

Overall, the source apportionment results based on LDI-MS data provide source separations with similar temporal behaviors as offline AMS and online AMS and ACSM analysis, yet seem to overestimate combustion related primary particle sources and underestimate secondary OA.






### 3.4.4 Uncertainty of PMF results

The uncertainty estimate, $\sigma_{tot}$, includes both the statistical uncertainty ($\sigma_{bs}$) and the uncertainty arising from the intraday variability of the measurements ($\sigma_{intrad}$, see Fig. 8, Section 2.3.2 and SI). In order to assess the impact of the intraday variability consideration ($\sigma_{intrad}$) in estimating the uncertainty besides the statistical uncertainty obtained from the bootstrapping approach ($\sigma_{bs}$), we compared the ratios $\sigma_{bs}/\sigma_{tot}$ for the different factors: for traffic1 the ratio was 0.83, for traffic2 0.75, for BBeff 0.86, for BBineff1 0.80, for BBineff2 0.72, for LMW-OA 0.68, and for bio-OA 0.68. Thus, it was important to propagate $\sigma_{intrad}$. The relative uncertainties for the median factor concentrations ranged within 0.15 (for traffic1), 0.16 (bio-OA), 0.17 (LMW-OA), 0.18 (BBineff2), 0.20 (traffic2), 0.22 (BBeff, and 0.28 (BBineff1). Unlike the relative error of $\sigma_{tot}$ of traffic1 (0.63 at the 10$^{th}$ percentile concentration and 0.13 at the 90$^{th}$ percentile concentration), traffic2 (0.68 and 0.11), BBeff (0.55 and 0.18), BBineff1 (0.47 and 0.20), BBineff2 (0.37 and 0.14), and LMW-OA (0.27 and 0.16) the relative error of bio-OA (0.20 and 0.15) only depended weakly on the factor concentration.

Throughout the measurement campaign, subsets of filters were analyzed repeatedly in order to assess the repeatability over longer time periods (see details in Supplement, Fig. S10). Most factors did not show significant changes of the attributed concentration during the measurement campaign. However, BBeff showed decreasing and LMW-OA increasing concentrations as a function of the measurement time. This could suggest an uncertain separation of these 2 factors. However, for these long time delays the variability increased and only few samples were repeated with such long time delays. Furthermore, the intra-day variance largely explains the total variance (traffic1 97%, traffic2 94%, BBeff 85%, BBineff1 89%, BBineff2 82%, LMW-OA 79%, bio-OA 97%, details in Supplement Fig. S10).

### 3.4.5 Factor variabilities and contributions

The time series of all factors are shown for the 9 sites in Fig. 9 as relative contributions to OA and summarized as yearly averages in Table 1. In the yearly average for all sites, traffic1 contributes 7%, traffic2 12%, BBeff 7%, BBineff1 17%, BBineff2 32%, LMW-OA 21%, and bio-OA 5% to OM as measured by LDI-MS.

At the southern Alpine Swiss sites, the wood burning influenced categories (WB=BBeff+BBineff1+ BBineff2) contribute more than at the northern sites (70% vs 50%). Moreover, in winter WB explains 81% and 61% of OM in the south and the north, respectively. This difference is mostly caused by the higher relative contribution of BBineff1 (34% in the south vs 14% in the north) since BBineff2 (28% and 30%) and BBeff (7% and 5%) do not show strong geographical differences. The ratio BBineff1/BB (Fig. 10b) shows enhanced contributions of BBineff1 at the southern Alpine sites, especially during high pollution episodes in winter. This suggests different wood burning regimes in the 2 regions, as already discussed above.

After scaling of the mass spectra to OM, absolute factor concentrations have still to be interpreted with caution. One reason for this is that the relative response factors of the sources/factors are not known, another is that certain species (in analogy to the cooking-aerosol sample) may not be detected by LDI-MS if externally mixed. In support of the quantitative interpretation of our results, we note that the sum of the wood burning related LDI-MS factors correlates well with the offline AMS counterpart ($R^2_{LDI,oAMS}$ =0.82, described in Section 3.4.3). On the yearly average 24% of the measured OM is apportioned to secondary OA. In summer, a bigger fraction is attributed to secondary OA (35% compared to 16% in winter). 35% (summer) and 1% (winter) of the secondary OA is attributed to biogenic sources. For some samples from the same period in Magadino, also the fossil and non-fossil content of OC and EC was determined (Vlachou et al., in prep) based on the method of Zhang et al. (2012). Vlachou et al. (in prep.) observed increased $OC_f$ /$EC_f$ ratios in summer which suggests other fossil POA sources in summer than in winter, or secondary formation of fossil OA. The higher traffic1 / $eBC_{tr}$ and traffic1/$EC_f$ ratios in




summer are in agreement with enhanced $OC_f/EC_f$ ratios (Fig. 10a). However, a part of traffic can also be mixed into BB leading to an underestimation of the traffic concentrations in winter. Overall this suggests that traffic1 represents aged or secondary traffic OA.

**4. Summary and conclusion**

In this study, we developed a novel method for the chemical characterization of particulate matter collected on quartz-fiber filters by LDI-MS. We applied the method to 819 samples. The method included the use of silver nitrate for $m/z$ calibration and the automated peak integration of the mass spectra at unit-mass resolution. The benefit of LDI-MS measurements for the chemical characterization and a better understanding of the sources contributing to the ambient $PM_{10}$ was assessed at nine

sites in central Europe throughout the entire year 2013.

Wood combustion smog chamber experiments revealed an influence of the burning conditions on the mass spectral signature. Tunnel samples used as a reference for traffic related emissions show mass spectral signatures distinctly different from wood combustion. Key $m/z$'s identified in the wood burning and traffic signatures showed links to expected markers as

e.g., levoglucosan and $NO_x$, respectively. The ambient mass spectral information was further used for source apportionment by PMF. Thereby, the influence of efficient and inefficient wood burning was separated. The extracted wood burning emissions correlated with the results from offline AMS source apportionment. Other components are more difficult to compare quantitatively because of different source separations in PMF as well as differences in the response factors of OA components. The influence of traffic emissions was represented by 2 factors. One of these could clearly be linked to BC-

related traffic ($eBC_{tr}$) and $NO_x$, and thus to primary emissions. The other, when normalized to $eBC_{tr}$, showed a similar behavior as $OC_f/EC_f$, and was therefore attributed to aged/secondary traffic OA. A factor was attributed to biogenic SOA based on its concentration exponentially increasing with temperature. Another OA factor was characterized by low-molecular-weight ions and was correlated with $NH_4^+$ and was attributed to SOA from an unknown source.

Acknowledgements
This work was supported by the Swiss Federal Office of Environment, Liechtenstein, Ostluft, the Cantons Basel, Graubünden, and Thurgau. We also thank AWEL Zurich for providing us with samples collected in Islisbergtunnel.

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



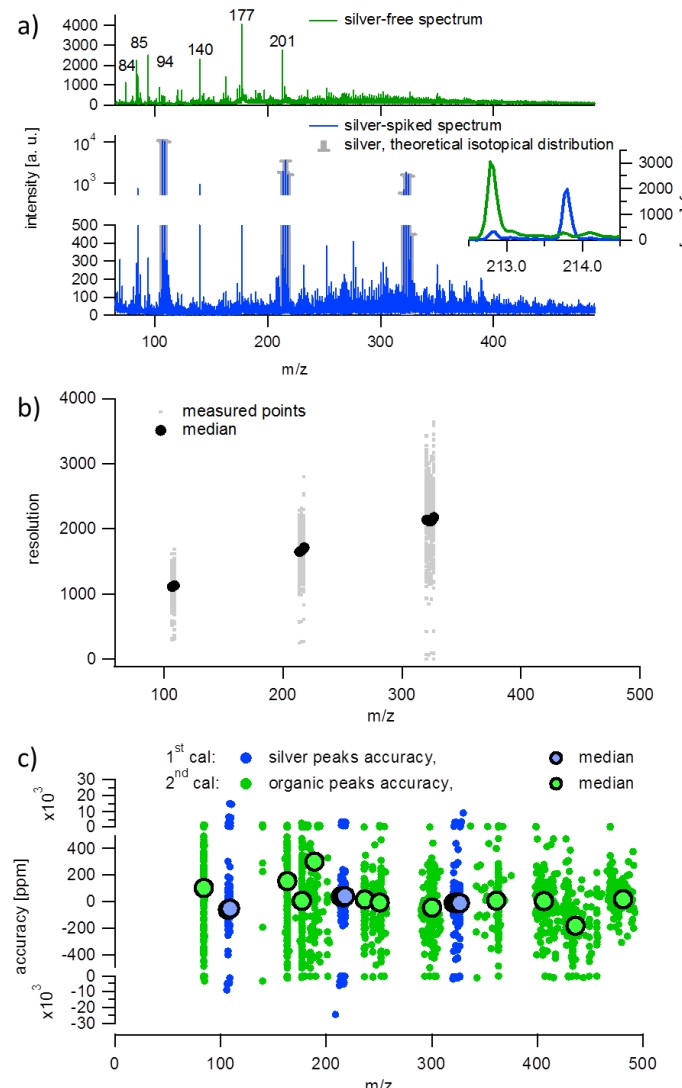

**Figure 1:** *m/z* **calibration of LDI-MS analysis of aerosol collected on quartz-fiber filter: a) example of calibrated silver (Ag-spec, blue) and no-silver-containing average mass spectra (noAg-spec, green) of a filter sample. The insert in a) displays a zoom-in of the Ag-spec and noAg-spec. b) operational resolution determined based on silver-mono-, di-, and trimer. c)** *m/z* **calibration accuracy for both steps of the** *m/z* **calibration.**





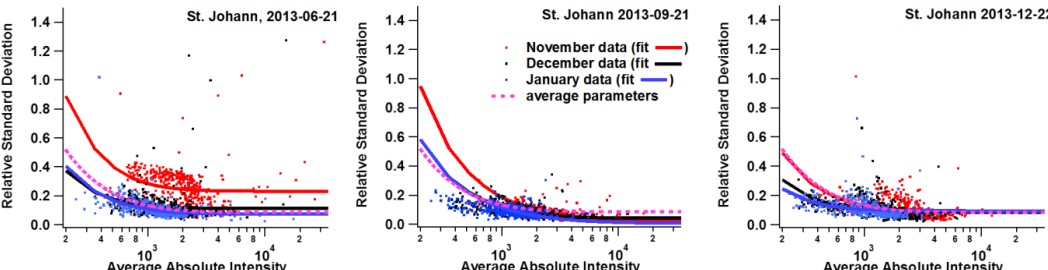

**Figure 2:** *Error model parametrization based on 3 samples (Basel, 2013-06-21, 2013-09-21, 2013-12-22) measured on 3 instances with each time 10 repeats on 1 sample holder.*

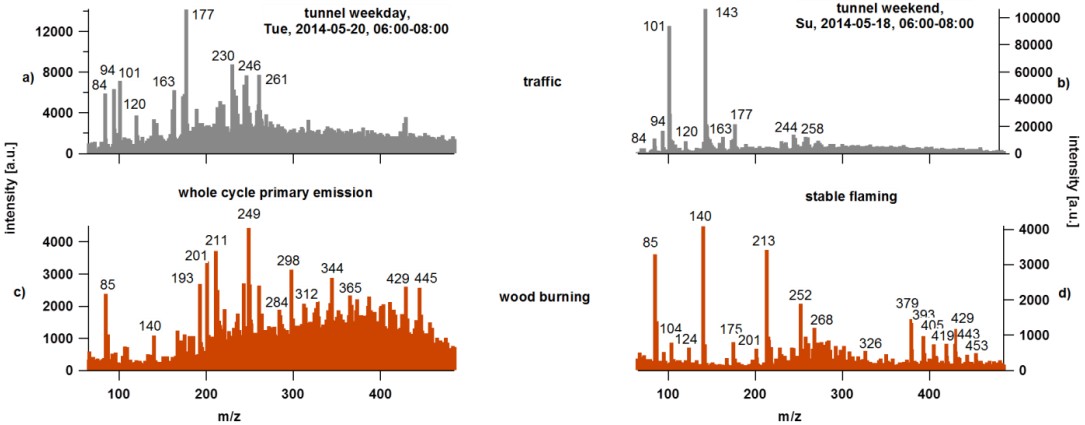

**Figure 3: samples representing different combustion sources a) and b) traffic emissions from samples collected in Islisberg tunnel**
10 **(exit, Wettswil, Switzerland), and c) primary wood burning using whole cycle emissions d) and stable flaming phase emissions d).**
**In absence of a measurable mass spectrum, no spectrum for cooking emissions is displayed.**





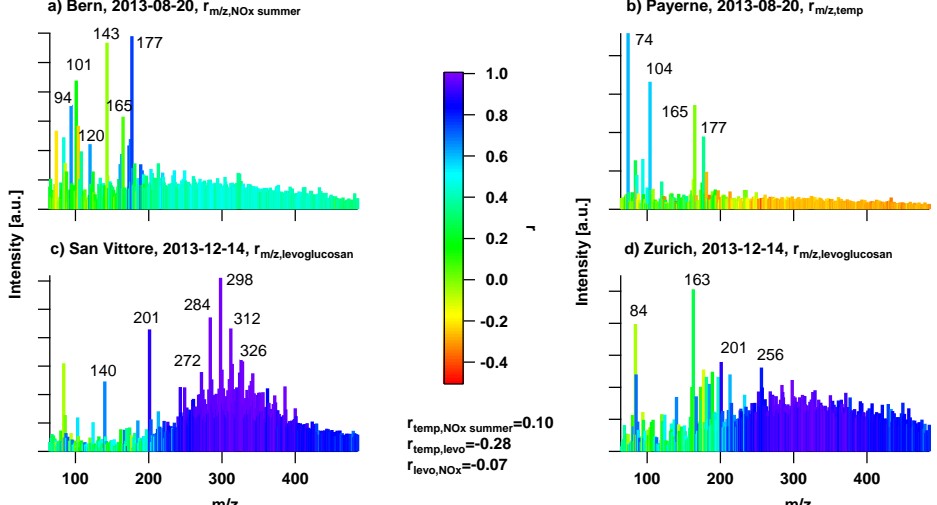

**Figure 4:** LDI-MS mass spectra from summer and winter samples (dates given in the legend) from a traffic-influenced (Bern), a rural (Payerne), a wood burning-influenced (San Vittore) and an urban background (Zurich) site. Spectra are color-coded with the correlation coefficient (r) between the *m/z*s and a specific environmental parameter for the whole dataset: for a) the correlation with $NO_x$ during summer, for b) the correlation with temperature, for c) and d) the correlation with levoglucosan.



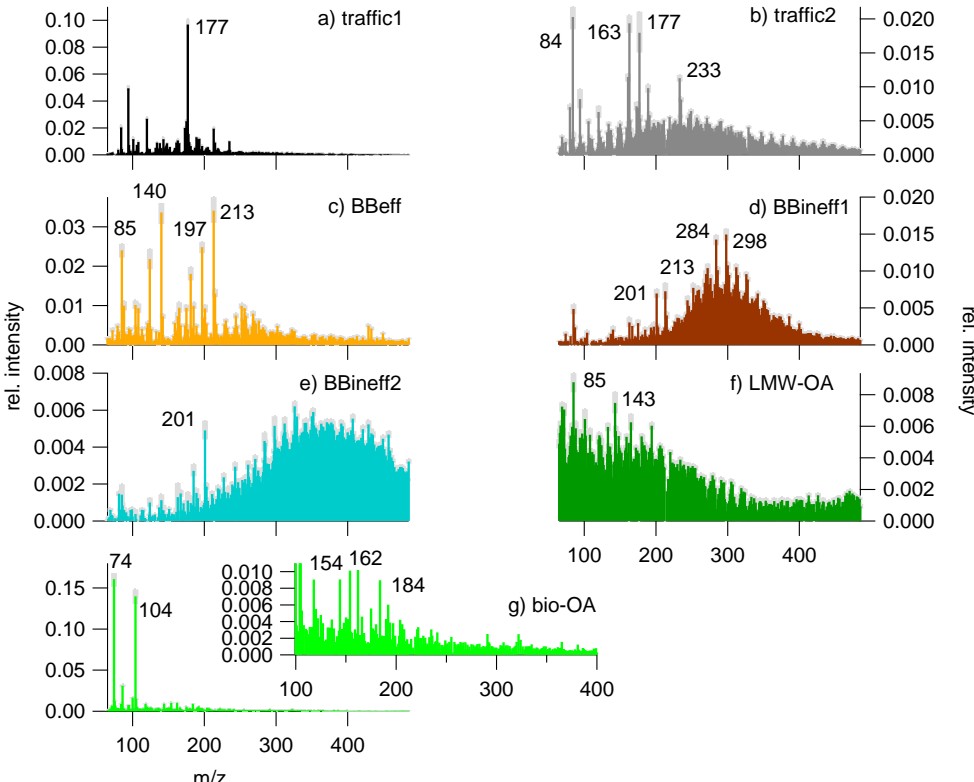

**Figure 5: PMF factor profiles (colored sticks) and their uncertainty (grey shaded areas, variability among PMF runs): a) traffic1, b) traffic2, c) efficient wood burning (BBeff), d) inefficient wood burning (BBineff1), e) inefficient wood burning 2 (BBineff2), f) lower molecular weight OA (LMW-OA), and g) biogenic OA (bio-OA).**





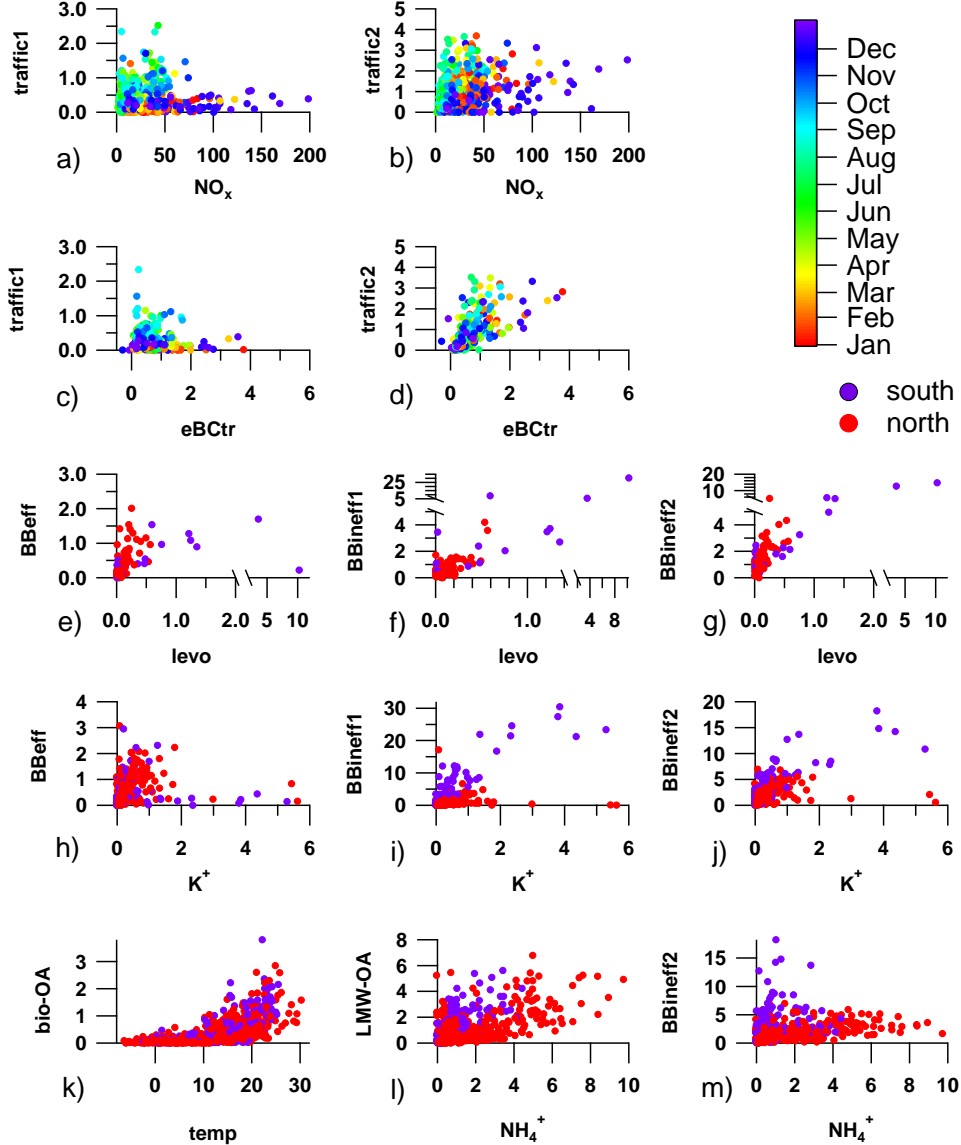

**Figure 6: Scatterplots between factor-time series and respective markers (traffic1, traffic2, BBeff, BBineff1, BBineff2, bio-OA, LMW-OA, eBC$_{tr}$, levoglucosan, potassium, ammonium are displayed in µg/m$^3$, NO$_x$ in ppm, and temperature in °C).**





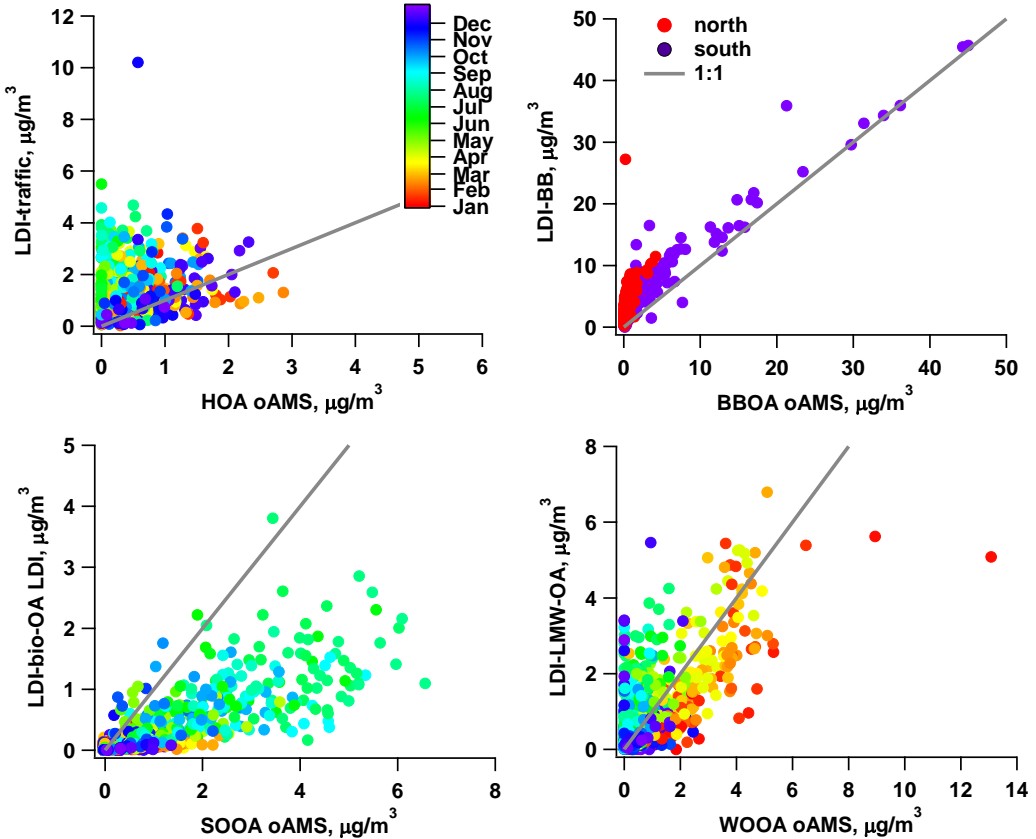

**Figure 7: Comparison of LDI-MS to reference offline AMS source apportionment results for the sum of traffic related factors (LDI-traffic, $R^2_{LDI,oAMS}$=0.05), sum of wood burning related factors (LDI-BB, $R^2_{LDI,oAMS}$ =0.82), bio-OA ($R^2_{LDI,oAMS}$ =0.61), and LMW-OA ($R^2_{LDI,oAMS}$ =0.47).**


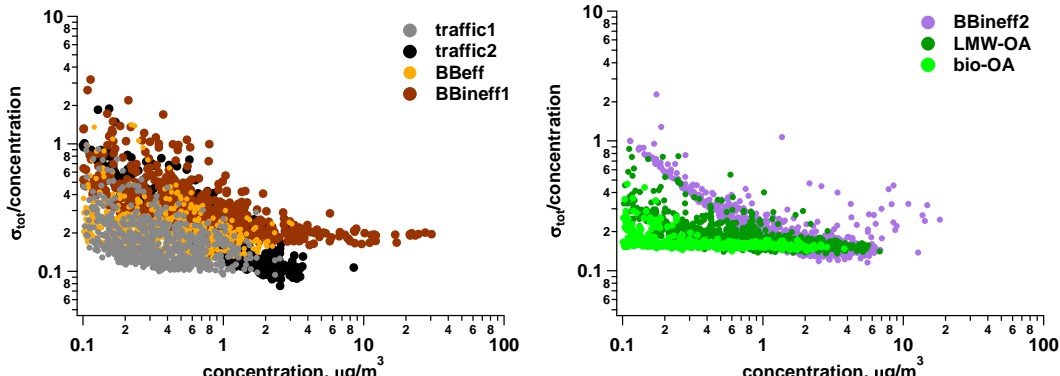

**Figure 8: Relative $\sigma_{tot}$ for the different PMF factors as a function of the factor concentration.**

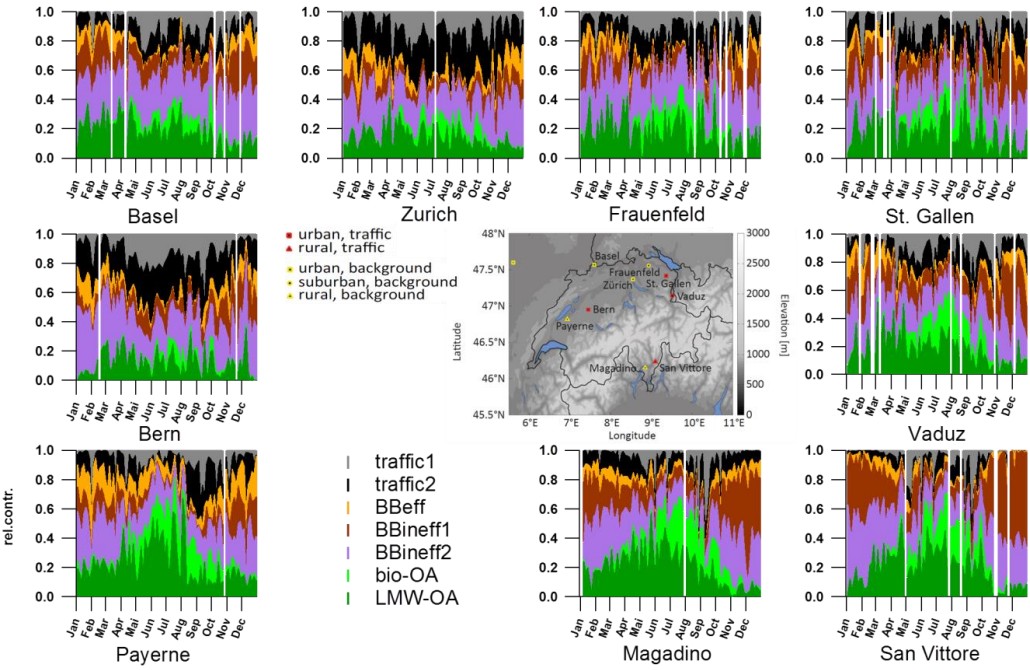

**Figure 9: Relative factor time series of 7 identified factors for all nine sites in study area.**




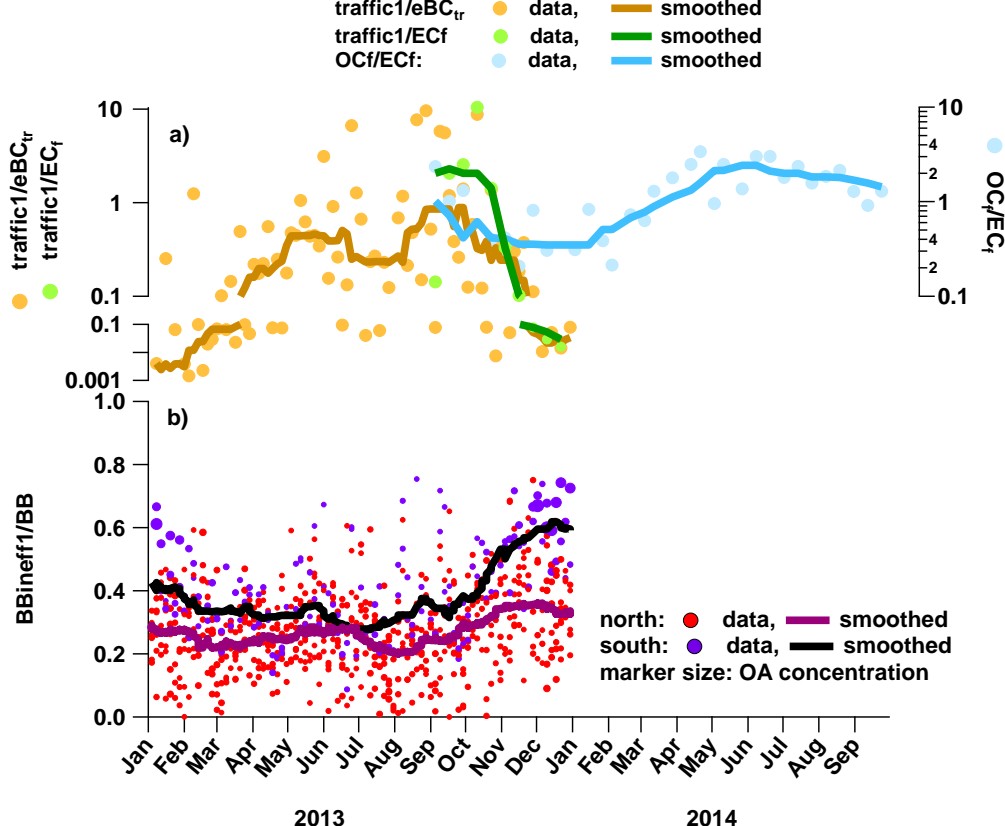

**Figure 10: a) time series of traffic1 normalized to eBC$_{tr}$ and EC$_f$ in comparison to OC$_f$/EC$_f$ in Magadino, b) influence of inefficient wood burning emissions (BBineff1) in comparison to the sum of wood burning influenced factors (BB=BBineff1+BBineff2+BBeff) for the entire datasets.**





**Table 1: yearly averages of the relative factor contributions.**

| yearly average, % | traffic1 | traffic2 | BBeff | BBineff1 | BBineff2 | bio-OA | LMW-OA |
|---|---|---|---|---|---|---|---|
| Basel | 6 | 12 | 7 | 17 | 32 | 5 | 21 |
| Bern | 11 | 24 | 7 | 13 | 30 | 3 | 12 |
| Frauenfeld | 8 | 16 | 6 | 13 | 30 | 7 | 20 |
| St. Gallen | 8 | 16 | 4 | 17 | 27 | 7 | 21 |
| Magadino | 3 | 11 | 7 | 27 | 27 | 8 | 17 |
| Payerne | 5 | 15 | 12 | 9 | 25 | 11 | 23 |
| Vaduz | 7 | 12 | 7 | 20 | 27 | 7 | 20 |
| S. Vittore | 2 | 4 | 3 | 40 | 32 | 5 | 14 |
| Zurich | 9 | 27 | 8 | 9 | 27 | 5 | 15 |