# Peer review of "Insights into organic-aerosol sources via a novel laser-desorption/ionization mass spectrometry technique applied to one year of $PM_{10}$ samples from nine sites in central Europe"

_Atmospheric Chemistry and Physics, 2017_

## Referee Comment (RC1) · Anonymous Referee #1 · 10 Sep 2017

Organic aerosol (OA) is a major component of the total aerosol. Thus, understanding the sources of OA is crucial for controlling aerosol pollution. In the past decade, the development of the aerosol mass spectrometer allows the direct quantification of organic aerosol mass, and source apportionment can be done by further applying PMF on such data set. The main weakness of AMS is that the hard ionization technique causes significant fragmentation, so that some sources are difficult to separate, such as secondary formation from anthropogenic or biogenic origins. To overcome this drawback, the authors, for the first time, explored the usage of LDI technique on offline samples, which causes less fragmentation and may give new insights of OA sources. The authors developed novel procedures for instrument calibration and uncertainty assessment, generated reference spectra for PMF from both specific environment (in the tunnel for traffic sources) and the lab (for wood burning sources), and carefully evaluated the results by comparing to the results from AMS technique. By doing so, the authors reported both the advantages and some potential issues of this technique, which makes this work easy to understand and follow. Although the ionization efficiency and the "matrix effect" of LDI technique remains poorly understood, which weakened the interpretability of some PMF factors, this work provided several new insights in OA sources and should be published in ACP after the following comments are addressed.

Minor comments:

Page 4, Line 16-17. "The observed ions are typically still fragments...". Some of the spectra (e.g. Fig.3d) reported in this work contains some high-mass peaks, which might be complete molecules. This work will be strengthened if you can identify them.

Page 5, Line 5-20 and Page 7, Line 5-10. The usage of AgNO3 seems unnecessary to me. First, using UMR data does not require a high-quality mass calibration. Second, you mentioned that AgNO3 may cause change in ionization physics, which might even hurt the mass calibration. If there are other important reasons for using AgNO3, please specify. Otherwise, I think it is important to mention that spiking AgNO3 should be skipped in future works.

Following this comment, the mass calibration accuracy (Fig.3c) is very unstable and too low under the mass resolution of 1000~2000. Do you have explanations?

Page 5, Line 31. Remove "scales"

Page 5, Line 34. It is the first time you mention "WSOC", please mention the full name.

Page5, Eq1. Scaling LDI intensity to AMS intensity may cause problems. In later anal-

Interactive
comment

yses, we know that LDI detection has bias on different OA sources, but AMS detection should be rather equal. If the fraction of different OA sources changed, for example, LDI-insensitive source increased but LDI-sensitive source decreased in the same magnitude, this change will be clearly seen as a decrease in LDI data, but NOT in AMS data. Under such circumstances, the scaling factor has a dependency on source distribution. This might (NOT necessarily) affect the overall PMF results, for example the seasonal variation of PMF factors and the comparison of LDI-PMF and AMS-PMF. Thus, I think it is necessary to double check the results using unscaled LDI data.

Page 7, Line 22-25. The fitting parameter in November is clearly different (Fig.2 red dots), though maybe not significantly.

Page 8, Line 33. ". . . of which 84, 94, 120, and 177 correlated with NOx. . .". 84amu is not marked in Fig.4a, and it does not show a strong correlation with NOx (R ≈ 0.2 as I read from the figure)

Page 9, Line 15. "Both traffic1 and traffic2 showed a relationship to NOx, only traffic2 correlated with eBCtr (Table S1)". What does the "relationship" mean? The correlation between traffic 1 and NOx is very weak. From Table S1, I think the traffic 1 has neither correlation with NOx, nor with eBC; while traffic 2 may correlate with both.

Page 9, Line 16-17. "This suggests that traffic2 is related to primary emissions from the combustion process and that traffic2 is also influenced by other processes, like e.g. secondary production." For people who are not familiar with traffic tracers, e.g. NOx and eBC, this interpretation is hard to follow. It is good to explain this a bit more.

Page 10, Line 31-33. "The contribution of the last remaining factor showed an exponential increase with temperature, similar to terpene emissions and biogenic SOA (Leaitch et al., 2011), suggesting this factor to be strongly influenced by biogenic SOA production. Therefore, this factor was termed bio-OA." Some other atmospheric processes, other than VOC emissions, are temperature-dependent, such as VOC oxidation and photochemistry. Is there additional evidence to validate the bio-OA factor, such as

higher fractional contribution in rural sites than in urban sites?

---

## Referee Comment (RC2) · Anonymous Referee #2 · 30 Oct 2017

Mass spectrometrical methods are in widespread use to characterize the organic composition of particulate matter. Aerosol mass spectrometers, despite their drawbacks with respect to hard ionization, are often drawn upon for source apportionment studies in combination with PMF. Here, the authors applied a LDI technique for an extensive study of ambient aerosol samples. Measurements and data analysis have been conducted meticulously, also reflecting on uncertainties, and source apportionment results have been compared to AMS data. The work should be published in ACP after some

minor revisions.

1) There is not much detail on the experimental setup of the mass spectrometrical experiment. Some hints are given, such as "ion extraction is not orthogonal to the ablation plume", then how is it performed?, or information about mass resolution. In the introduction, the adjustment of laser power is mentioned as one of the advantages of LDI-MS compared to ATOF-MS, but information on the laser power in this experiment is missing save the fact that it is adjustable. Therefore, the authors are encouraged to give more details on the laser/mass spectrometer system.

2) The mass resolution is low in this experiment, so it is understandable that the authors did not assign their signals to actual compounds. But I wonder, is there not the possibility to assign at least some prominent peaks relating to past experience or literature data? In this respect, there are many odd mass numbers in the spectra, hinting at the formation of fragment ions. This should be discussed briefly.

3) Figure 3 reveals, if there are a lot of signals resulting from a sample, a big unresolved hump is visible ( as can be seen from the wood burning in comparison to the clean, well structured spectrum from tunnel weekend). Can this be improved/targeted by reducing the laser power, or will this lead to a severe and unacceptable loss in sensitivity?

4) What is the gain in spiking with silver nitrate? In the end, you have to calibrate the spectra without showing peaks from silver ions with an independent calibration procedure, as is mentioned in the manuscript. Why then not apply the second procedure, for which obviously no spiking is necessary, in general and omit the silver nitrate solution?

5) What is the difference between WSOM and WSOC?

6) Page 2, line 2: Omit the impact with EI, call it just electron ionization.

7) Please use "such as" instead of "like" in elaborating insertions.

8) Page 8, line 28: Please do not use Thompson as unit of m/z differences. Maybe Dalton or stay with m/z.
* * *
9) Page 13, line 6: In the first sentence of the conclusion, I would not call it a novel method, rather the utilization of a known method in use since approx. 2000 with some new developed novel aspects.

---

## Author Response (AR1)

We thank the referees for their comments, which helped improving the quality of our manuscript. A point by point response (in black) to the reviewers' comments (in blue) will follow. Changes in the text are indicated in in *black italics*.

Reviewer 1:

Organic aerosol (OA) is a major component of the total aerosol. Thus, understanding the sources of OA is crucial for controlling aerosol pollution. In the past decade, the development of the aerosol mass spectrometer allows the direct quantification of organic aerosol mass, and source apportionment can be done by further applying PMF on such data set. The main weakness of AMS is that the hard ionization technique causes significant fragmentation, so that some sources are difficult to separate, such as secondary formation from anthropogenic or biogenic origins. To overcome this drawback, the authors, for the first time, explored the usage of LDI technique on offline samples, which causes less fragmentation and may give new insights of OA sources. The authors developed novel procedures for instrument calibration and uncertainty assessment, generated reference spectra for PMF from both specific environment (in the tunnel for traffic sources) and the lab (for wood burning sources), and carefully evaluated the results by comparing to the results from AMS technique. By doing so, the authors reported both the advantages and some potential issues of this technique, which makes this work easy to understand and follow. Although the ionization efficiency and the "matrix effect" of LDI technique remains poorly understood, which weakened the interpretability of some PMF factors, this work provided several new insights in OA sources and should be published in ACP after the following comments are addressed.

Minor comments:

Page 4, Line 16-17. "The observed ions are typically still fragments: : :". Some of the spectra (e.g. Fig.3d) reported in this work contains some high-mass peaks, which might be complete molecules. This work will be strengthened if you can identify them.

We agree with the reviewer that the identification of ions in the recorded mass spectra would greatly help gaining a deeper understanding of sources and processes but also of the measurement technique itself. In the complex mixtures that we analyze, different ions have plausibly a significant contribution to the signal at a single nominal mass, especially given the size of the ions that we observe. With the mass resolution and mass accuracy of the m/z calibration an identification of the contributions of different ions is not possible unambiguously. Stark et al. (2015) developed an approach to estimate the carbon oxidation state using the mass-to-charge ratio of a peak and its mass defect (difference to closest integer) for mass spectra with limited mass resolution. With our instrumental approach a similar approach cannot be applied because of the poor mass accuracy of the m/z calibration.

We have adapted the text according to the reviewer's suggestion:

*"… With this accuracy and resolution, neither distinguishing different ions at the same nominal mass nor estimating properties such as O/C and H/C for a nominal mass following Stark et al. (2016) was possible for the quartz-filter-LDI-MS measurements. Instead, the spectra were integrated to UMR sticks. …"*

As reviewer 2 mentions, the prominent presence of odd masses might hint to a rather prominent fraction of signal attributable to fragments as reviewer 2 mentioned. However, it is currently impossible to distinguish odd mass molecular contributors from fragments. According to Samburova et al. (2005a), fragmentation in this type of instrument is negligible. The mass spectral signatures acquired in this study are similar to the ones in Samburova et al. (2005a) and the lower laser energy applied in this study is lower than in the former study. This suggests that also during our measurements fragmentation was not a prominent process.

We added this discussion to the manuscript:

"…
*The presence of considerable signal at odd masses might indicate that significant fragmentation occurs during desorption and ionization of the organic matter in the LDI-MS (Fig. 1a, 3, 4). However, Samburova et al. (2005a) suggested that fragmentation in this instrument is negligible. The mass spectral signatures acquired in this study are similar to the ones in Samburova et al. (2005a) and the lower laser energy applied in this study is lower than in the formed study. This suggests that also during our measurements fragmentation was not a prominent process. Overall, the extent of fragmentation remains unclear.*
*…"*

Page 5, Line 5-20 and Page 7, Line 5-10. The usage of $AgNO_3$ seems unnecessary to me. First, using UMR data does not require a high-quality mass calibration. Second, you mentioned that $AgNO_3$ may cause change in ionization physics, which might even hurt the mass calibration. If there are other important reasons for using $AgNO_3$, please specify. Otherwise, I think it is important to mention that spiking $AgNO_3$ should be skipped in future works. Following this comment, the mass calibration accuracy (Fig.3c) is very unstable and too low under the mass resolution of 1000-2000. Do you have explanations?

The m/z calibration parameters in our LDI-MS measurements were highly variable between different samples. Therefore, to determine unit mass resolution integration regions it was necessary to perform an m/z calibration on every single sample.

In other mass spectrometers, some peaks are always observed, independent of the sample, and can therefore be used for an m/z calibration and/or to assess the performance of an initial m/z calibration: E.g. in the Aerodyne AMS the ions $N_2^+$, $CO_2^+$, $W^+$, etc. are present under most conditions. As another example, in chemical ionization mass spectrometry m/z calibration is typically performed using the known chemical reagent ions. For a reliable m/z calibration such known anchor ions are crucial. In many complex mixtures as for ambient aerosol, the signal observed at any nominal mass is composed of ions from numerous organic compounds and peaks are broad. Therefore, the exact mass of such a peak is unknown hindering an accurate m/z calibration.

In matrix assisted LDI-MS matrix molecules or matrix fragment ions have been used previously. However, the native matrix of our samples was highly variable, so that no ions provided suitable anchors for calibration. Therefore, we chose to spike our samples with $AgNO_3$. In the absence of such anchor ions an m/z calibration, even to integer accuracy, is not feasible.

Indeed, as mentioned in the reviewer's comment, we observed differences between the average spectrum containing $AgNO_3$ and the average spectrum without $AgNO_3$ which we related to changing ionization conditions. Hence, we did not report atmospheric data for $AgNO_3$-spiked samples. Rather, we used the $AgNO_3$-spiked spectra to obtain a first calibration (based on $Ag^+$ anchor ions), used the first calibrate to define prominent, filter-specific anchor ions, and then calibrated the unspiked spectra using these anchor ions. This

procedure is explained in Section 2.2.2 Data treatment of the revised manuscript (unchanged from original) and was essential to perform an objective calibration. Because only prominent peaks were used in the second calibration step, the possibility of an influence of $AgNO_3$ on these samples is minimal.

Although we had mentioned these issues in the submitted manuscript, we have clarified them further following both reviewers' comments. We adapted the respective part of the manuscript to:

*"…*
*The m/z calibration parameters were highly variable between different samples. Therefore, to determine unit mass resolution integration regions it was necessary to perform an m/z calibration on every single sample. However, unlike in the Aerodyne AMS ($N_2^+$, $O_2^+$, $W^+$), there are no dominant anchor ions present in these spectra that could be used for an m/z calibration. In absence of such ions, we performed a two-step calibration procedure. Each sample was spiked with silver nitrate ($AgNO_3$, Sigma Aldrich, >99.8%) as an internal standard (approach illustrated in Fig. 1). In order to avoid the suppression of the sample signal, the internal standard (aqueous solution 500 ppt to 20 ppm) was only placed (as a droplet) on a small part of the sample. The 499 spectra from all positions on the measurement grid were separated and defined as (1) silver-spiked (lower panel in Fig. 1a), (2) silver-free (upper panel in Fig. 1a), and (3) intermediate-silver. Intermediate-silver cases were defined using the signal intensity in the regions of the mass spectrum where silver was expected, in comparison to adjacent silver-free regions of the spectrum, and were discarded. To calculate the first calibration, we calibrated the average silver-spiked spectrum of each filter sample, using the peaks of the silver monomer (m/z 107, 109 Th), dimer (m/z 214, 216, 218 Th), and trimer (m/z 321, 323, 325, 327 Th). We found that this calibration of silver-spiked was not directly applicable to the average silver-free mass spectrum. Possibly, spiking the filter region with aqueous $AgNO_3$ caused enough of a change to the surface of the filter sample to influence the ionization physics. This could affect the m/z calibration, since the delayed-pulse ion extraction in our instrument is not orthogonal to the ablation plume but nearly parallel. Therefore, a second calibration was obtained for the averaged silver-free spectra using prominent non-silver peaks present in both spectra. Such a two-step calibration is necessary to achieve accurate m/z calibrations of the silver-free spectra.*
*…"*

The mass resolution as presented in Fig. 1b is computed based on the silver peaks since all other peaks are potentially influenced by a multitude of ions. Mass resolution and accuracy are independent properties; broad peaks can still be accurately calibrated in the absence of interference. We acknowledge that the mass accuracy in our dataset is poor. Therefore, we emphasize in the text that based on such accuracy an identification of contributing ions to a nominal mass is not possible (i.e. high-resolution analysis).

*"… With this accuracy and resolution, neither distinguishing different ions at the same nominal mass nor estimating properties such as O/C and H/C for a nominal mass following Stark et al. (2016) was possible for the quartz-filter-LDI-MS measurements. Instead, the spectra were integrated to UMR sticks.…"*

Since the ions are not orthogonally extracted into the ToF unit, variability in the ionization process might be propagated into the ToF unit and, thereby, contribute to the poor accuracy of the m/z calibration. Since this variability was observed for the 1[st] calibration, which used only the Ag "anchor" ions which dominated the spectral intensity (so that a substantial interference by unidentified ions is very unlikely), it must be generally attributed to variability in the ionization and detection processes.

An observation of calibration inaccuracy in the calibration itself indicates that the calibration model was imprecise. The calibration model is the standard equation describing ion time-of-flight in the mass spectrometer, and would be violated if ions were at different distances from the detector or had different kinetic energy vectors, upon extraction. A non-uniform electric field in the spectrometer could also cause aberrances. These problems have been minimized in some instruments, but their minimization would reduce our ability to directly analyze filter samples, which is a major strength of our approach.

Page 5, Line 31. Remove "scales"

We removed this word.

Page 5, Line 34. It is the first time you mention "WSOC", please mention the full name.

We added this information:

*"… OC$_{Sunset}$ was determined using the Sunset OC/EC analyzer and (WSOM/WSOC)$_{oAMS}$ through WSOM measurements (described in Section 2.1, WSOC being water-soluble organic carbon). …"*

Page5, Eq1. Scaling LDI intensity to AMS intensity may cause problems. In later analyses, we know that LDI detection has bias on different OA sources, but AMS detection should be rather equal. If the fraction of different OA sources changed, for example, LDI-insensitive source increased but LDI-sensitive source decreased in the same magnitude, this change will be clearly seen as a decrease in LDI data, but NOT in AMS data. Under such circumstances, the scaling factor has a dependency on source distribution. This might (NOT necessarily) affect the overall PMF results, for example the seasonal variation of PMF factors and the comparison of LDI-PMF and AMS-PMF. Thus, I think it is necessary to double check the results using unscaled LDI data.

The reviewer addresses one of the main challenges related to LDI-MS analyses: the quantification. During our experiments we discovered that (1) the signal recorded for the same sample during 1 day is fairly variable introducing a lot of variability and (2) that instrumental changes over time (possibly properties of the laser) introduced a decrease of signal with the measurement time. Consequently, the raw signal does not show a relation with OC, the sum of OC and EC or PM10 and is, therefore, not interpretable (Fig. S3, S4). Therefore, scaling to an external property is inevitable.

We updated the offline AMS source apportionment results used in the comparison (Fig 7) with the recently published data in Daellenbach et al. (2017). In the same time we also reworked Fig. 9 and 10 for an improved visibility. This resulted in small changes in Fig. 7 and the related text:

*"…*

*As was the case for the comparison to NO$_x$ and eBC$_{tr}$ (Section 3.4.2), also in the comparison to the offline AMS traffic (HOA), a higher LDI-traffic/HOA was observed in summer which contributes to the low correlation coefficient (R$^2$=0.04). This might be related to the fact that HOA is only primary*

*without major contributions of traffic SOA or dust resuspension while LDI-traffic is potentially also influenced by aged/secondary traffic aerosol and resuspension. In contrast to summer, we observe a good agreement between LDI-traffic and HOA in winter. Thus even though a varying relative response factor (rRF) of the LDI-MS might contribute to these differences, these biases are not systematic but season dependent. As stated earlier, LDI-traffic is thought to be a mixture of primary tailpipe exhaust, aged/secondary tailpipe exhaust, and resuspended dust (as well as tyre break and engine wear). This will be further elucidated in Section 3.4.5.*

*LDI-BB was highly correlated with offline AMS-BBOA ($R^2$=0.83), yet the LDI-BB concentrations were higher than BBOA from oAMS, especially in northern Switzerland (with an LDI-BB:AMS-BBOA ratio between 1.4 and 4.1 for the different sites). A possible reason is also here the mixing of secondary components into LDI-BB when comparing to primary BBOA from oAMS. However, we cannot exclude that this effect is due to different rRFs of the LDI-MS for different compound classes.*

*The identified secondary components, LMW-OA ($R^2$=0.45) and bio-OA ($R^2$=0.62), also correlate with the corresponding OOA factors from the oAMS analysis, WOOA and SOOA, yet the correlation coefficients are smaller than for LDI-BB and BBOA. For these factors, differences between the two methodologies could be related to differences in the PMF performance or to differences in the response factors for different components in the LDI-MS (the LMW-OA: WOOA ratio is between 0.7 and 1.1 and the bio-OA: SOOA ratio between 0.2 and 0.4).*

*…"*

[Figure]

**Figure 1: Comparison of LDI-MS to reference offline AMS source apportionment results for the sum of traffic related factors (LDI-traffic, $R^2_{LDI,oAMS}$=0.04), sum of wood burning related factors (LDI-BB, $R^2_{LDI,oAMS}$ =0.83), bio-OA ($R^2_{LDI,oAMS}$ =0.62), and LMW-OA ($R^2_{LDI,oAMS}$ =0.45).**

In response to the question, we added a paragraph on the analysis of relative response factors for LDI-MS in Section 3.4.3 of the manuscript:

*"…*

*Overall, the source apportionment results based on LDI-MS data provide source separations with similar temporal behaviors as offline AMS and online AMS and ACSM analyses, yet seem to overestimate combustion related primary particle sources and underestimate secondary OA. However, we assume for this comparison that all factors give an equal response at a given concentration, i.e. the relative response factor (rRF$_{LDI}$) of all factors to be 1. The relative contribution of a factor k to the total signal observed with the LDI-MS $(r_{i,k,LDI})$ a specific measurement (i) depends on the factor concentration $(g_{i,k,LDI})$ as well as the sum of all factors $(\sum_k^p(g_{i,k,LDI}))$ separated for the LDI-MS data. Assuming that differences between the LDI-PMF and the AMS-PMF arise solely from different rRF$_{LDI}$s for different factors, $r_{i,k,LDI}$ is a function of the AMS factor concentrations $(g_{i,k,AMS})$ and the rRF$_{LDI}$ of this factor:*

$$r_{i,k,LDI} = \frac{g_{i,k,LDI}}{\sum_k^p (g_{i,k,LDI})} = \frac{g_{i,k,AMS} * rRF_{k,LDI}}{\sum_k^p (g_{i,k,AMS} * rRF_{k,LDI})} \qquad (4)$$

*In order to determine $rRF_{LDI}$ for the LDI-MS several strong assumptions are required: (1) the sum of traffic1 and traffic2 represents HOA, (2) the sum of WBeff, WBineff1 and WBineff2 represents BBOA, (3) LMW-OA represents WOOA, (4) bio-OA represents SOOA and (5) the AMS factors for which there is an LDI-MS equivalent, (1) - (4), are the only contributors to OA. In scenario 1 All the above assumptions are considered to be true. For scenario 1, we estimate $rRF_{LDI-BB}$ as 1.84, $rRF_{LDI-LMW-OA}$ as 0.29 and $rRF_{LDI-bio-OA}$ as 0.40 (using LDI-traffic as reference factor, $rRF_{LDI-traffic}$ = 1.00). The LDI-MS factor concentrations corrected using $rRF_{LDI}$ show a close relation to the uncorrected LDI factor concentrations ($R^2_{LDI-traffic}$=0.78, $R^2_{LDI-BB}$=0.94, $R^2_{LDI-LMW-OA}$=0.80, $R^2_{LDI-bio-OA}$=0.85, Fig. 8).*

*We tested the sensitivity of the $rRF_{LDI}$ estimates to the assumptions (1) and (2) being wrong (scenario 2) and only assumption (5) being wrong (scenario 3). In scenario 2, we alter scenario 1 by comparing the sum of traffic1 and bio-OA to SOOA and the sum of BBineff2 and LMW-OA to WOOA when computing $rRF_{LDI}$. For scenario 2, LDI-MS factor concentrations corrected using $rRF_{LDI}$ show also a close relation to uncorrected LDI-MS factor concentrations ($R^2_{LDI-traffic2}$=0.96, $R^2_{BBeff+BBineff1}$=1.00, $R^2_{LDI-BBineff2+LMW-OA}$=0.99, $R^2_{traffic1+bio-OA}$=0.96). In scenario 3, we alter scenario 1 by considering also AMS factors without an equivalent in the LDI-MS PMF. Therefore, we compare LDI-traffic to the sum of HOA, COA, and SC-OA. Under these conditions LDI-MS factor concentrations corrected using $rRF_{LDI}$ show a close relation to uncorrected LDI-MS factor concentrations ($R^2_{LDI-traffic}$=0.75, $R^2_{LDI-BB}$=0.90, $R^2_{LMW-OA}$=0.74, $R^2_{bio-OA}$=0.85). The differences in $rRF_{LDI}$ between scenarios 1, 2, and 3 highlight the uncertainties caused by plausible violations of the underlying assumptions when determining $rRF_{LDI}$. Overall the estimates $rRF_{LDI}$ are highly uncertain and their accurate determination needs to be the focus of future work. Given the high uncertainty of $rRF_{LDI}$ and the good correlation between $rRF_{LDI}$ corrected and uncorrected LDI-MS factor concentrations, we present uncorrected results without considering $rRF_{LDI}$.*

**Table 1: relative response factors (*rRF*) for LDI-MS analyses for 3 different scenarios.**

| scenario 1 | scenario 2 | scenario 3 |
|---|---|---|
| $rRF_{LDI-traffic}$ = 1.00±0.00 | $rRF_{LDI-traffic2}$ = 1.00±0.00 | $rRF_{LDI-traffic}$ = 1.00±0.00 |
| $rRF_{LDI-BB}$ = 1.84±0.06 | $rRF_{LDI-BBeff+LDI-BBineff1}$ = 1.26±0.06 | $rRF_{LDI-BB}$ = 5.67±0.13 |
| $rRF_{LDI-LMW-OA}$ = 0.29±0.02 | $rRF_{LDI-LMW-OA+LDI-BBineff2}$ = 1.67 ±0.07 | $rRF_{LDI-LMW-OA}$ = 0.30±0.02 |
| $rRF_{LDI-bio-OA}$ = 0.14±0.01 | $rRF_{LDI-bio-OA+LDI-traffic1}$ = 0.39±0.02 | $rRF_{LDI-bio-OA}$ = 0.65±0.02 |

[Figure]

**Figure 2: Comparison of LDI-MS factor concentrations corrected using relative response factors (*rRF*) to uncorrected LDI factor concentrations (Scenario 1).**

*…"*

Additionally, we also add a comment in the conclusion:

*"… Other components are more difficult to compare quantitatively because of different source separations in PMF as well as differences in the relative response factors (rRF) of OA components. rRF determined in this study are uncertain and, therefore, not used for correcting the LDI-PMF results. …"*

Page 7, Line 22-25. The fitting parameter in November is clearly different (Fig.2 red dots), though maybe not significantly.

The purpose of these exercises was to obtain an error model as presented in Section 3.1 and in the Supplement. There are only three repeats. Therefore, it is hard to judge whether the experiment in question is an outlier or the result of a broad distribution. We report the resulting uncertainty of the error model in the manuscript as following:

*"… The relative error was related to the absolute signal intensity for each m/z for each sample and experiment (Fig. 2), with an asymptotic value of only ~9 ± 7 % at high signals (average and standard deviation of fitting Eq. 3). The average absolute error for small signals was 102 ± 48 a.u. (from Eq. 3)…"*

Page 8, Line 33. ": : : of which 84, 94, 120, and 177 correlated with NOx…". 84amu is not marked in Fig.4a, and it does not show a strong correlation with NOx (R ~0.2 as I read from the figure)

The sentence has been corrected:

*"… The m/z 84 94, 101, 120, 143, 165, and 177 Th, also detected in tunnel samples, showed high relative intensities in Bern in summer, of which 94, 120, and 177 Th correlated with $NO_x$, which suggests a high contribution of traffic emissions at this location, consistent with previous observations (Zotter et al., 2014). …"*

Page 9, Line 15. "Both traffic1 and traffic2 showed a relationship to NOx, only traffic2 correlated with eBCtr (Table S1)". What does the "relationship" mean? The correlation between traffic 1 and NOx is very weak. From Table S1, I think the traffic 1 has neither correlation with NOx, nor with eBC; while traffic 2 may correlate with both.

We rephrased the respective part of the manuscript:

*"… Both traffic1 and traffic2 showed increasing concentrations with increasing $NO_x$ levels. However traffic1/NOx and traffic2/NOx were seasonally variable ($R_{p,traffic1,NOx}$=-0.17, $R_{p,traffic2,NOx}$=0.24, Fig. 6, Fig. S9). While traffic2 correlated with $eBC_{tr}$ , traffic1 did not show such a dependency (Tab. S1, Fig. 6, Fig. S9). …"*

Page 9, Line 16-17. "This suggests that traffic2 is related to primary emissions from the combustion process and that traffic2 is also influenced by other processes, like e.g. secondary production." For people who are not familiar with traffic tracers, e.g. NOx and eBC, this interpretation is hard to follow. It is good to explain this a bit more.

We rephrased the respective part of the manuscript:

*"… The correlation with $eBC_{tr}$ suggests that traffic2 is related to primary emissions from the combustion process. Based on the lack of correlation between traffic1 and $eBC_{tr}$, traffic1 might also be influenced by other processes, such as e.g. secondary production. …"*

Page 10, Line 31-33. "The contribution of the last remaining factor showed an exponential increase with temperature, similar to terpene emissions and biogenic SOA (Leaitch et al., 2011), suggesting this factor to be strongly influenced by biogenic SOA production. Therefore, this factor was termed bio-OA." Some other atmospheric processes, other than VOC emissions, are temperature-dependent, such as VOC oxidation and photochemistry. Is there additional evidence to validate the bio-OA factor, such as higher fractional contribution in rural sites than in urban sites?

Using yearly average NOx concentrations as a metric for rurality/urbanity, Payerne sticks out as rural (low NOx) and Bern as urban (high NOx). In Payerne bio-OA has a higher relative contribution (11%) than in Bern (3%) which are the highest and lowest in the entire dataset (see Table 2 in manuscript). Qualitatively, the same was observed in the AMS data (38% SOOA in Payerne and 21%; only S. Vittore showed a lower contribution at the expense of higher BBOA, Daellenbach et al., 2017).

In order to make this information more visible we added yearly NOx concentrations to Tab. 2 and sorted the sites according to the yearly average NOx concentration.

Such information was added to the manuscript:

*"…Bio-OA had a highest relative contribution at the most rural site in the dataset (Payerne: yearly average bio-OA 11%, yearly average NOx concentration: 9 ppb) and the lowest at the most trafficated site (Zurich: yearly average bio-OA 3%, yearly average NOx concentration: 48 ppb). …"*

In response to this question, we added the further information on the laser power and ion extraction to the following paragraphs and adapted the sentence on the comparison between ATOFMS and LDI-MS:

*"… Possibly, spiking the filter region with aqueous AgNO₃ caused enough of a change to the surface of the filter sample to influence the ionization physics. This could affect the m/z calibration, since ion extraction in our instrument is not orthogonal to the ablation plume but a delayed pulsed extraction. …"*

*"…*
*We recorded the mass spectra of 819 filter samples in an m/z range 65-500 thomsons (Th; 1 Th =1 Da e⁻¹, where e is the elementary charge, ion gate at m/z 60 Th) using a laser-desorption/ionization-ToF MS (Shimadzu Axima Confidence, Shimadzu-Biotech Corp., Kyoto, Japan) equipped with an N₂ laser (wavelength 337 nm, frequency 50 Hz, laser pulse width 3 ns, 130-180 μJ/pulse) in the positive reflectron mode. All the accessible instrumental parameters were kept constant during the whole period of measurements taking place from November 2015 to mid-March 2016. Specifically, the laser intensity was adjustable by means of a rotating wheel of filters with varying transmissivity (0 being blocked and 180 being completely open). We set the wheel parameter to 105 of 180 which would result in an estimated laser energy of ~6-9 μJ/pulse or 2.8-4.2\*10⁸ W/cm² with a 3ns pulse and 30μm laser beam diameter. While the laser energy was initially set and kept constant, the aging of the laser during the given time period was also expected to reduce its intensity. We monitored and assessed changes in laser power and other instrumental parameters, as well as possible sources of uncertainty/contamination from sample preparation and intra and inter-day reproducibility, by repeated measurements of a subset of our samples.*
*…"*

*"… Similar to the AMS, online single-particle LDI-MS instruments such as the ATOFMS also yield extensive fragmentation. However, such fragmentation can be avoided by measuring offline aerosol samples (filters) using other systems. Samburova et al. (2005a) showed by comparing*

*measurement with and without matrix addition, that fragmentation was negligible in their instrument (wavelength 337 nm, LDI-MS, Shimadzu/Kratos, Axima CFR). ..."*

2) The mass resolution is low in this experiment, so it is understandable that the authors did not assign their signals to actual compounds. But I wonder, is there not the possibility to assign at least some prominent peaks relating to past experience or literature data? In this respect, there are many odd mass numbers in the spectra, hinting at the formation of fragment ions. This should be discussed briefly.

As the reviewer points out it is impossible to distinguish the contribution of different ions at the same nominal mass. We agree with the reviewer that it is desirable to assign compounds to the most prominent masses and that the prominent presence of odd masses hints to a rather prominent fraction of signal attributable to fragments. However, it is currently impossible to distinguish odd mass molecular contributors from fragments.

In absence of vast literature databases with the same instrument, we prefer to refrain from interpreting the fingerprints further than by comparing to reference samples since we fear to over-interpret the data. Earlier work confirms the prevalence of the same peaks for 1 site present in our dataset but cannot attribute these peaks to specific compounds (Zurich, Kaserne, Samburova et al., 2005).

As reviewer 2 mentions, the prominent presence of odd masses might hint to a rather prominent fraction of signal attributable to fragments as reviewer 2 mentioned. However, it is currently impossible to distinguish odd mass molecular contributors from fragments. According to Samburova et al. (2005a), fragmentation in this type of instrument is negligible. The mass spectral signatures acquired in this study are similar to the ones in Samburova et al. (2005a) and the lower laser energy applied in this study is lower than in the former study. This suggests that also during our measurements fragmentation was not a prominent process. We mention the possible influence of fragmentation now also in the main text:

*"...*
*The operational mass resolution, m/Δm, based on the measurements conducted on one sample tray per month and the accuracy of both calibration steps are shown in Fig. 1b and 1c, respectively. The operational resolution of the instrument was only determined for the silver mono- (resolution 1100), di- (1700), and trimer (2100) as we are confident of the absence of interfering ions for these peaks. In comparison to the LDI-MS, the HR-ToF AMS in V-mode (and W-mode) has a resolution of ca. 2'000 (4'000) at m/z 100 Th and an m/z calibration accuracy < 20 ppm (< 10 ppm, DeCarlo et al., 2006). With this accuracy and resolution, neither distinguishing different ions at the same nominal mass nor estimating properties such as O/C and H/C for a nominal mass following Stark et al. (2016) was possible for the quartz-filter-LDI-MS measurements. Instead, the spectra were integrated to UMR sticks. The presence of considerable signal at odd masses might indicate that significant fragmentation occurs during desorption and ionization of the organic matter in the LDI-MS (Fig. 1a, 3, 4). However, Samburova et al. (2005a) suggest that fragmentation in this instrument is negligible. The mass spectral signatures acquired in this study are similar to the ones in Samburova et al. (2005a) and the lower laser energy applied in this study is lower than in the formed study. This suggests that also during our measurements fragmentation was not a prominent process. Overall, the extent of fragmentation remains unclear.*

*..."*

3) Figure 3 reveals, if there are a lot of signals resulting from a sample, a big unresolved hump is visible ( as can be seen from the wood burning in comparison to the clean, well

structured spectrum from tunnel weekend). Can this be improved/targeted by reducing the laser power, or will this lead to a severe and unacceptable loss in sensitivity?

For such separations, we suggest to extract PM with different solvents in order to get a better idea on their properties in future work. Samburova et al. (2005) observed such humps also in water-soluble PM for 1 of the sites present also in this study (Zurich, Kaserne).

4) What is the gain in spiking with silver nitrate? In the end, you have to calibrate the spectra without showing peaks from silver ions with an independent calibration procedure, as is mentioned in the manuscript. Why then not apply the second procedure, for which obviously no spiking is necessary, in general and omit the silver nitrate solution?

Both reviewers mentioned a similar point. We have taken this feedback as an opportunity to better clarify our two-step calibration procedure, as detailed more fully in the following (the same response has been written to both reviewers):

The m/z calibration parameters in our LDI-MS measurements were highly variable between different samples. Therefore, to determine unit mass resolution integration regions it was necessary to perform an m/z calibration on every single sample.

In other mass spectrometers, some peaks are always observed, independent of the sample, and can therefore be used for an m/z calibration and/or to assess the performance of an initial m/z calibration: E.g. in the Aerodyne AMS the ions $N_2^+$, $CO_2^+$, $W^+$, etc. are present under most conditions. As another example, In chemical ionization mass spectrometry m/z calibration is typically performed using the known chemical reagent ions. For a reliable m/z calibration such known anchor ions are crucial. In many complex mixtures as for ambient aerosol, the signal observed at any nominal mass is composed of ions from numerous organic compounds and peaks are broad. Therefore, the exact mass of such a peak is unknown hindering an accurate m/z calibration.

In matrix assisted LDI-MS matrix molecules or matrix fragment ions have been used previously. However, the native matrix of our samples was highly variable, so that no ions provided suitable anchors for calibration. Therefore, we chose to spike our samples with $AgNO_3$. In the absence of such anchor ions an m/z calibration, even to integer accuracy, is not feasible.

Indeed, as mentioned in the reviewer's comment, we observed differences between the average spectrum containing $AgNO_3$ and the average spectrum without $AgNO_3$ which we related to changing ionization conditions. Hence, we did not report atmospheric data for $AgNO_3$-spiked samples. Rather, we used the $AgNO_3$-spiked spectra to obtain a first calibration (based on $Ag^+$ anchor ions), used the first calibrate to define prominent, filter-specific anchor ions, then calibrated the unspiked spectra using these anchor ions. This procedure is explained in Section 2.2.2 Data treatment of the revised manuscript (unchanged from original) and was essential to enabling an objective calibration to be performed. Because only prominent peaks were used in the second calibration step, the possibility of an influence of $AgNO_3$ on these samples is minimal.

Although we had mentioned these issues in the submitted manuscript, we have clarified them further following both reviewers' comments. We adapted the respective part of the manuscript to:

[revised manuscript text omitted]

8) Page 8, line 28: Please do not use Thompson as unit of m/z differences. Maybe Dalton or stay with m/z.

We prefer to use Thomson as a unit for m/z for the case that some ions have more than one charge. We now include a statement on the unit of m/z that we use in this manuscript, including a description on how Thomson (Th) relate to Dalton (Th=Da e$^{-1}$). Additionally, we also added the unit of m/z throughout the manuscript wherever it was missing.

*"... We recorded the mass spectra of 819 filter samples in an m/z range 65-500 thomsons (Th; 1 Th =1 Da e$^{-1}$, where e is the elementary charge, ion gate at m/z 60 Th) using a laser-desorption/ionization-ToF MS (Shimadzu Axima Confidence, Shimadzu-Biotech Corp., Kyoto, Japan) equipped with an N2 laser (wavelength 337 nm, frequency 50 Hz, laser pulse width 3 ns, 130-180 µJ/pulse) in the positive reflectron mode. ..."*

*"... In general, the mass spectra from winter-time were characterized by higher contributions from higher-molecular-weight fragments compared to summer. For the winter sample from San Vittore, a large "hump" was present, on top of which signals appeared with high intensities and a highly regular pattern of m/z differences of 14 Th. ..."*

9) Page 13, line 6: In the first sentence of the conclusion, I would not call it a novel method, rather the utilization of a known method in use since approx. 2000 with some new developed novel aspects.

We changed the manuscript according to the reviewer's suggestion.

*"... In this study, we advanced a known method for the chemical characterization of particulate matter collected on quartz-fiber filters by LDI-MS and applied the method to 819 samples. ..."*

[revised manuscript text omitted]
} = 1.00\pm0.00$ | $rRF_{LDI\text{-}traffic2} = 1.00\pm0.00$ | $rRF_{LDI\text{-}traffic} = 1.00\pm0.00$ |
| $rRF_{LDI\text{-}BB} = 1.84\pm0.06$ | $rRF_{LDI\text{-}BBeff+LDI\text{-}BBineff1} = 1.26\pm0.06$ | $rRF_{LDI\text{-}BB} = 5.67\pm0.13$ |
| $rRF_{LDI\text{-}LMW\text{-}OA} = 0.29\pm0.02$ | $rRF_{LDI\text{-}LMW\text{-}OA+LDI\text{-}BBineff2} = 1.67\pm0.07$ | $rRF_{LDI\text{-}LMW\text{-}OA} = 0.30\pm0.02$ |
| $rRF_{LDI\text{-}bio\text{-}OA} = 0.14\pm0.01$ | $rRF_{LDI\text{-}bio\text{-}OA+LDI\text{-}traffic1} = 0.39\pm0.02$ | $rRF_{LDI\text{-}bio\text{-}OA} = 0.65\pm0.02$ |

[Figure]

**Figure 8: Comparison of LDI-MS factor concentrations corrected using relative response factors (*rRF*) to uncorrected LDI factor concentrations (Scenario 1).**

[Figure]

**Figure 9:** Relative $\sigma_{tot}$ for the different PMF factors as a function of the factor concentration.

[Figure]

Figure 9

Field Code Changed

[Figure]

**Figure 10**: Relative factor time series of 7 identified factors for all nine sites in study area.

[Figure]

**Figure 11: a) time series of traffic1 normalized to eBC$_{tr}$ and EC$_f$ in comparison to OC$_f$ /EC$_f$ in Magadino, b) influence of inefficient wood burning emissions (BBineff1) in comparison to the sum of wood burning influenced factors (BB=BBineff1+BBineff2+BBeff) for the entire datasets.**

Table 21: yearly averages of the relative factor contributions and NOx concentrations.

| yearly average, % | traffic1 | traffic 2 | BBeff | BBineff1 | BBineff2 | bio-OA | LMW-OA | NOx, ppb |
|---|---|---|---|---|---|---|---|---|
| BernBasel | 611 | 1224 | 7 | 1713 | 3230 | 53 | 2112 | 48 |
| ZurichBern | 119 | 2427 | 78 | 139 | 3027 | 35 | 1215 | 24 |
| St. GallenFrauenfeld | 8 | 16 | 64 | 1317 | 3027 | 7 | 2021 | 24 |
| BaselSt. Gallen | 86 | 1612 | 47 | 17 | 2732 | 75 | 21 | 21 |
| FrauenfeldMagadino | 38 | 1116 | 76 | 2713 | 2730 | 87 | 1720 | 21 |
| VaduzPayerne | 57 15 | 12 | 97 | 2520 | 1127 | 237 | 20 | 20 |
| PayerneVaduz | 75 | 15 | 12 | 79 | 2025 | 2711 | 723 | 209 |
| MagadinoS. Vittore | 23 | 411 | 37 | 4027 | 3227 | 58 | 1417 | 20 |
| S. VittoreZurich | 92 | 274 | 83 | 940 | 2732 | 5 | 1514 | 18 |